# Group-robust Machine Unlearning

**Thomas De Min**
*University of Trento, Italy*
*thomas.demin@unitn.it*

**Subhankar Roy**
*University of Bergamo, Italy*

**Stéphane Lathuilière**
*Inria Grenoble, Univ. Grenoble Alpes, France*

**Elisa Ricci**
*University of Trento, Italy*
*Fondazione Bruno Kessler, Italy*

**Massimiliano Mancini**
*University of Trento, Italy*

**Reviewed on OpenReview:** *https://openreview.net/forum?id=StSq7mpUVw*

## Abstract

Machine unlearning is an emerging paradigm to remove the influence of specific training data (i.e., the forget set) from a model while preserving its knowledge of the rest of the data (i.e., the remaining set). Previous approaches assume the forget data to be uniformly distributed from all training datapoints. However, if the data to unlearn is dominant in one *group* (e.g., ethnicity, gender), we empirically show that performance for this group can degrade, leading to fairness issues. To perform unlearning while preserving fairness, this work addresses the overlooked problem of non-uniformly distributed forget sets, which we refer to as *group-robust machine unlearning*. We formalize the problem and present a simple and effective exact unlearning strategy that mitigates the performance loss in dominant groups via sample distribution reweighting. Moreover, we present MIU (Mutual Information-aware Machine Unlearning), the first approach for group robustness in approximate machine unlearning. MIU minimizes the mutual information between model features and group information, achieving unlearning while reducing performance degradation in the dominant group of the forget set. Additionally, MIU exploits sample distribution reweighting and mutual information calibration with the original model to preserve group robustness. We conduct experiments on three datasets and show that MIU outperforms standard methods, achieving unlearning without compromising model robustness. Source code available at `https://github.com/tdemin16/group-robust_machine_unlearning`.

## 1 Introduction

In several countries, recent regulations grant users greater control over their digital privacy, explicitly recognizing their right to request the removal of personal data, known as the "right to be forgotten" (Mantelero, 2013; Voigt & Von dem Bussche, 2017). Therefore, to comply with such regulations, machine learning systems should be able to forget specific training data upon user request—a challenge addressed by machine unlearning (Kurmanji et al., 2024; Chundawat et al., 2023a).

In the machine unlearning literature, the standard setup assumes that a group of individuals requests the removal of their data. A machine unlearning algorithm then fulfills this request by making the model "forget"

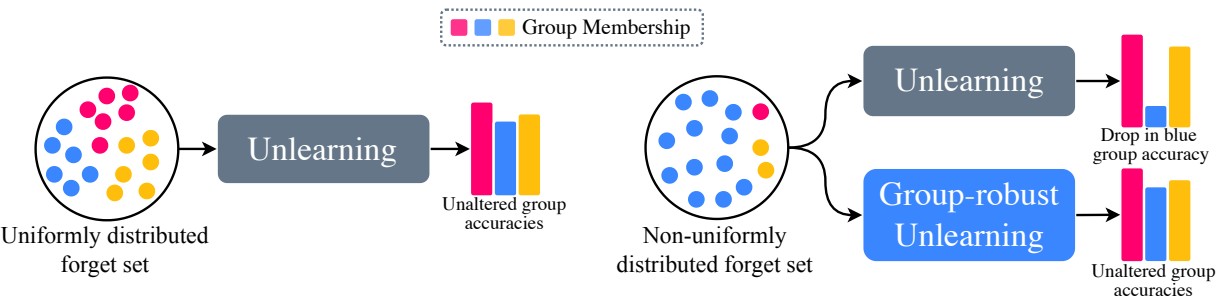

Figure 1: **Comparing unlearning approaches.** Previous works assume the forget set to be uniformly distributed. However, real-life unlearning requests do not comply with the uniform distribution assumption (Bertram et al., 2019). If the forget set distribution is predominant in some groups (e.g., old males), it can lead to performance degradation in such dominant forget groups (i.e., the blue group in the figure). Group-robust Unlearning prevents this from happening.

the designated data (i.e., the forget set) while maintaining its utility (Kurmanji et al., 2024). The naive solution, called *exact unlearning*, consists of retraining the model without the forget data; however, this is computationally prohibitive for large models (Zhao et al., 2024b). *Approximate unlearning* methods (Jia et al., 2023; Fan et al., 2024b; Kurmanji et al., 2024) overcome this limitation by unlearning the model with significantly fewer resources but with less unlearning guarantees.

Most approaches assume that the forget sets are uniformly sampled from the training data (Jia et al., 2023; Fan et al., 2024b; Cheng & Amiri, 2023; Shen et al., 2024; He et al., 2024; Zhao et al., 2024b; Huang et al., 2024). However, studies (Bertram et al., 2019; Zhang et al., 2024) show that individuals from different social and cultural backgrounds (i.e., *groups*) request to be forgotten at varying rates: Bertram et al. (2019) reports that 44.4% of unlearning requests to news websites relate to professional wrongdoing or crimes, while Zhang et al. (2024) finds that wealthier, highly educated individuals are more likely to request unlearning. Existing methods overlook this imbalance, potentially degrading the accuracy of dominant groups in the forget set and leading to unfair outcomes (see Figs. 1 and 2). This can be critical in scenarios requiring high accuracy across all groups. In a recommendation system, if unlearning requests predominantly come from a specific group (e.g., young males), the system's quality for that group may degrade, potentially making it unusable. Thus, machine unlearning algorithms must account for distribution shifts.

We target this unexplored scenario, which we name *group-robust machine unlearning*, that aims to unlearn the user's data while minimizing the performance deterioration of groups dominating the forget set. Differently from previous works, we tackle both *exact* and *approximate* unlearning by (i) finding a retraining strategy that preserves the original group robustness (Sagawa et al., 2020), and (ii) proposing an approximate unlearning method that efficiently forgets data while preserving robustness for unbalanced forget sets.

For (i), we show that reweighting the sampling distribution during retraining compensates for information loss with minimal robustness impact. We validate this strategy (called REWEIGHT) against GROUP-DRO (Sagawa et al., 2020), a popular group-robust optimization method, showing that REWEIGHT better preserves model group robustness in exact unlearning. For (ii), we introduce an approximate unlearning method called MIU (Mutual Information-aware Machine Unlearning), leveraging mutual information minimization (Belghazi et al., 2018) and calibration to unlearn the forget set while preserving model robustness. By minimizing the mutual information between forget-set features and ground-truth group annotations, we decorrelate unlearning from spurious attributes (Liu et al., 2021), mitigating performance loss for dominant groups. To prevent affecting other groups, we calibrate the unlearned model's mutual information to match the original one. Coupled with REWEIGHT, MIU outperforms established unlearning approaches (L1-SPARSE (Jia et al., 2023), SALUN (Fan et al., 2024b), and SCRUB (Kurmanji et al., 2024)) on CelebA (Liu et al.,

2015), Waterbirds (Sagawa et al., 2020), and FairFace (Karkkainen & Joo, 2021) in both unlearning efficacy and preserved group robustness.[1]

**Contributions.** In summary, our contributions are:

- We are the first to identify the issue of group robustness in approximate unlearning, showing how existing unlearning algorithms degrade model robustness in this setting.

- We propose a simple and effective sample distribution reweighting strategy to mitigate the group accuracy degradation in exact unlearning.

- We introduce MIU, the first approximate unlearning approach tailored for this task that minimizes the mutual information on the forget set while calibrating it to match the group robustness of the original model.

- We benchmark existing baselines and MIU on group-robust machine unlearning using CelebA (Liu et al., 2015), Waterbirds (Sagawa et al., 2020), and FairFace (Karkkainen & Joo, 2021), and showing that MIU outperforms existing methods in this task.

## 2 Related work

**Machine unlearning.** Exact machine unlearning methods (Bourtoule et al., 2021; Yan et al., 2022; Aldaghri et al., 2021) guarantee that sensitive data is removed. However, they are impractical (Chundawat et al., 2023a; Kurmanji et al., 2024) as retraining (part of) the model is prohibitive (Nguyen et al., 2022). Instead, approximate unlearning concentrates on computational feasibility by relaxing the guarantees constraints (Kurmanji et al., 2024; Chundawat et al., 2023a; Jia et al., 2023; Fan et al., 2024b; Chen et al., 2023). Most works focus on random (Jia et al., 2023; Fan et al., 2024b; Cheng & Amiri, 2023; Shen et al., 2024; He et al., 2024; Zhao et al., 2024b; Huang et al., 2024), and class (Chen et al., 2023; Chundawat et al., 2023a;b; Hoang et al., 2024; Cheng et al., 2024; Zhao et al., 2024a) unlearning, respectively forgetting i.i.d. data points, and all data points from a single class. While the former lowers the forget-set accuracy to match that of the model retrained without the unlearning data, the latter aims at scoring zero accuracy on the unlearned class. A few works (Chundawat et al., 2023a; Foster et al., 2024; Cheng et al., 2024) explore subclass unlearning, where a subset of a class (e.g., subclass *car* from superclass *vehicles*) is removed from model weights. While subclass and group-robust unlearning share similarities, the latter focuses on preserving group accuracies, whereas subclass unlearning alters the model to behave as if the target subclass was never in the training set. Related to our research, Fan et al. (2024a) suggests a bi-level optimization to sample adversarial forget sets that are difficult to unlearn. Also related, Chen et al. (2024) proposes machine unlearning as an efficient debiasing technique. Instead, Zhang et al. (2024) investigates how uniform and non-uniform data sampling affect fairness in MLPs and tabular data, limiting the evaluation to exact unlearning.

Group-robust unlearning can be seen as a generalized case of random unlearning, where we drop the i.i.d. assumption of the forget set. This has implications for the design of proposed methodologies. As existing approaches assume a uniform distribution of the forget set during evaluation, they do not account for group imbalance in the forget distribution, possibly degrading the model's robustness.

**Group-robust learning.** Methods for group-robust optimization train deep learning models to be robust to spurious correlations. Algorithms are categorized based on their access to group information. Within those that assume access to group annotations (Sagawa et al., 2020; Goel et al., 2020; Zhang et al., 2021; Idrissi et al., 2022; Kirichenko et al., 2023), group-DRO (Sagawa et al., 2020) dynamically reweights the misclassification penalty for each group to optimize the worst-group accuracy. Instead, Idrissi et al. (2022) proposes a simple baseline that subsamples each group to match the size of the smallest one. While these works usually achieve better results, accessing the group information can be challenging (e.g., annotation cost). Within methods agnostic to group information (Liu et al., 2021; Zhang et al., 2022; Idrissi et al., 2022; Sohoni et al., 2020; Nam et al., 2020), JTT (Liu et al., 2021) increases the sampling probability of wrongly classified data to improve worst-group accuracy. Correct-N-Contrast (Zhang et al., 2022) uses a contrastive

---

[1]We use robustness and group-robustness interchangeably in this paper.

loss to pull correctly and misclassified samples of the same class while pushing apart wrongly classified data points of different categories. However, methods that require group information also show state-of-the-art performance on groups discovered from data (Kim et al., 2024; D'Incà et al., 2024).

This paper is the first to study the intersection between group robustness and machine unlearning. For this reason, we assume complete access to the group information.

## 3 Method

This section formulates the machine unlearning task (Sec. 3.1) and the group-robust machine unlearning problem (Sec. 3.2). Then, it introduces the proposed sample distribution reweighting strategy (Sec. 3.3), showing its effectiveness in group-robust machine unlearning. Finally, Sec. 3.4 describes MIU, our approximate unlearning method tailored for group-robust unlearning.

### 3.1 Machine unlearning

Let $h_\varphi \circ f_\theta : X \to Y$ be a learnable function (or model), where $f_\theta(\cdot) : X \to Z$ is a non-linear feature extractor parameterized by $\theta$, and $h_\varphi(\cdot) : Z \to Y$ is a linear classifier parameterized by $\varphi$, mapping inputs from the image $X$ to the target space $Y$. Let $\mathcal{D}_{tr} = \{(\mathbf{x}, y)_i\}_{i=1}^{N_{tr}}$ be a training dataset of size $N_{tr}$, where $\mathbf{x}_i$ is an image, and $y_i$ its target label (e.g., age). Let a model trained on $\mathcal{D}_{tr}$ with algorithm $\mathcal{T}$ be denoted as $h_{\varphi_o} \circ f_{\theta_o}$ (or PRETRAIN). A machine unlearning algorithm $\mathcal{U}$ scrubs the influence of a desired forget set $\mathcal{D}_f \subset \mathcal{D}_{tr}$ from the pretrained model by outputting scrubbed weights $\{\varphi_u, \theta_u\}$, such that $h_{\varphi_u} \circ f_{\theta_u}$ is as close as possible to the exact unlearning model $h_{\varphi_r} \circ f_{\theta_r}$ (or RETRAIN), trained solely on the remaining set $\mathcal{D}_r = \mathcal{D}_{tr} \setminus \mathcal{D}_f$ with algorithm $\mathcal{T}$ (Kurmanji et al., 2024; Fan et al., 2024b; Zhao et al., 2024b).

### 3.2 Group-robust machine unlearning

In group-robust machine unlearning, we consider the forget data non-uniformly distributed. Therefore, let us re-define the training dataset as $\mathcal{D}_{tr} = \{(\mathbf{x}, y, a)_i\}_{i=1}^{N_{tr}}$, where $\mathbf{x}_i$ and $y_i$ are defined as in Sec. 3.1, and $a_i$ is the protected or sensitive attribute (e.g., *gender, ethnicity*). Now, let $G : Y \times A$ be the set of all groups, defined as the cartesian product between the target label set $Y$ and the protected attribute set $A$ (Sagawa et al., 2020; Kirichenko et al., 2023). We denote the *i-th* datapoint group as $g_i = (y_i, a_i)$. Each target-sensitive attribute pair (e.g., *males*, between the ages of *20-29*) identifies a unique group.[2]

If the forget set is non-i.i.d., the group distribution in the remaining set changes compared to the original training set. As the proportion of a group in the remaining set lowers, its resulting accuracy also decreases, which can harm the model's generalization performance on the dominant group of the forget

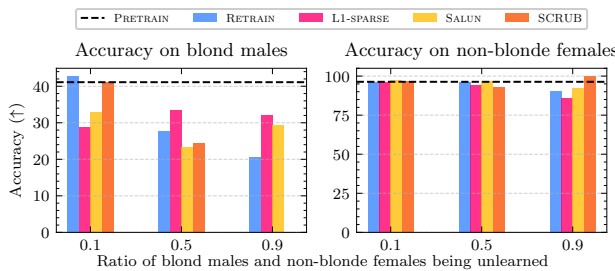

Figure 2: **Unlearning non-uniformly distributed data.** We test standard model retraining, and popular approximate unlearning methods (L1-SPARSE, SALUN, and SCRUB) in group-robust unlearning. The more samples of the least represented group (*blond males*) are unlearned from CelebA, the lower the model accuracy on that group. On the contrary, the most represented one (*non-blonde females*) is less affected.

set. Figure 2 shows the accuracy degradation caused by jointly removing different percentages of the least and most represented groups, i.e., *blond males* and *non-blonde females*. While the most represented group is mildly affected as its ratio drops from 44% to 7% (over four groups), the least represented one (dropping from 0.8% to 0.1%) suffers a substantial accuracy degradation of ∼20%. Further analysis is provided in Appx. B.

---

[2]*Target* and protected attributes are color-coded.

Therefore, the objective of group-robust machine unlearning differs from that of machine unlearning. While the latter minimizes the discrepancy with the retrained model (gold standard), the former also preserves the original model's performance on each group.

### 3.3 Frustratingly easy group-robust unlearning

Section 3.2 introduces the group-robust machine unlearning task and shows the extent to which the accuracy of the forget set dominant group(s) drops after unlearning. Unlike prior works (see Sec. 1), we aim at unlearning non-uniform forget data while preserving the model accuracy on the dominant group of the forget set. Therefore, this section proposes an exact unlearning strategy tailored to this task. To mitigate the performance degradation of the dominant group of $\mathcal{D}_f$, we argue that reweighting the data distribution (RWG in Idrissi et al. (2022)) to account for the removed samples is a simple and effective baseline that retrains the model with a minimal performance drop. Contrary to RWG (Idrissi et al., 2022), which enforces uniform sampling distribution from group-perspective, we suggest instead to reweight the sampling distribution to match the original dataset statistics. Intuitively, increasing the sampling likelihood for partly unlearned groups rebalances the remaining set group statistics to match those of the training dataset, recovering the original robustness.

Formally, let $P(\mathbf{x}_i) = \frac{1}{N_{tr}}$ be the probability of sampling $\mathbf{x}_i$, let $\nu_{tr}, \nu_r \in \mathbb{N}^G$ respectively be the group frequencies of the training and remaining datasets. We reweight $P(\mathbf{x}_i)$ according to the ratio $\alpha = \frac{\nu_{tr}}{\nu_r}$: $P(\mathbf{x}_i) = \frac{\alpha_{g_i}}{\sum_j \alpha_{g_j}}$, where $\alpha_{g_i}$ is the adjusted group weight for sample $\mathbf{x}_i$. We denote the data distribution reweighting strategy (or REWEIGHT) as $\omega(\cdot)$, i.e., $\omega(\mathcal{D}_r)$ denotes the reweighting strategy applied to the remaining dataset.

We validate REWEIGHT by comparing the model retrained with REWEIGHT and GROUP-DRO (Sagawa et al., 2020). Figure 3 summarizes the outcome of our analysis. As a byproduct of strongly optimizing worst-group accuracies, we notice that GROUP-DRO (Sagawa et al., 2020) can also increase the forget set accuracy. This issue makes approximate unlearning evaluation more difficult if RETRAIN + GROUP-DRO is used as the gold standard. Assuming a hypothetical original forget-set accuracy of 70%, if an approximate unlearning algorithm targeting such a gold standard leads to higher accuracy (e.g., 80%), then *was the knowledge unlearned if forget-set accuracy increased?* Answering this question is non-trivial, as the retrained model cannot be accessed for real applications. Furthermore, retraining with GROUP-DRO (Sagawa et al., 2020) causes a test accuracy drop (-5.8%) in FairFace (Karkkainen & Joo, 2021). On the contrary, REWEIGHT does not suffer from these issues.

### 3.4 Group-robust unlearning without retraining

Section 3.3 shows that sample reweighting is a simple and effective approach to machine unlearning that preserves the original model's robustness. However, model retraining is inefficient (Chundawat et al., 2023a; Kurmanji et al., 2024) and unpractical (Nguyen et al., 2022). Therefore, this section proposes MIU, our approximate machine unlearning algorithm that jointly tackles unlearning while decorrelating unlearning and group robustness without retraining the entire model.

As *scrubbing* unlearning data affects the forget set dominant group accuracy, we propose a unified objective for jointly unlearning while preserving original group robustness. We use mutual information between output features and group annotation, which, upon minimization on forget data, jointly un-

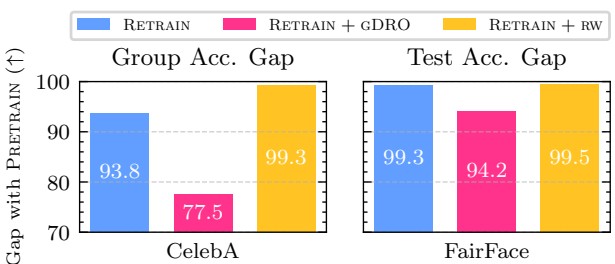

Figure 3: **REWEIGHT vs. GROUP-DRO.** RETRAIN + REWEIGHT achieves a better test and group accuracy alignment with the original model (higher is better). Thus, it better preserves the original performance after unlearning.

learns and mitigates the performance degradation on the dominant group of the forget set (Ragonesi et al.,

2021). Formally, let $I(Z; G) = I(Z; (Y, A))$ be the mutual information between random variables $Z$ and $G$, associated with model features $\mathbf{z} = f_\theta(\mathbf{x})$ and group $g = (y, a)$, then:

$$I(Z; G) = \int_G \int_Z \mathbb{P}_{(Z,G)}(\mathbf{z}, g) \log \left( \frac{\mathbb{P}_{(Z,G)}(\mathbf{z}, g)}{\mathbb{P}_Z(\mathbf{z})\mathbb{P}_G(g)} \right) d\mathbf{z} \, dg, \tag{1}$$

where $\mathbb{P}_{(Z,G)}$ is the joint pdf of $Z$ and $G$, and $\mathbb{P}_Z$, $\mathbb{P}_G$ are marginal pdfs of $Z$ and $G$. Minimizing $I(Z, G)$ reduces the dependency between $Z$ and $Y$ while also reducing it between $Z$ and $A$, jointly unlearning the network and disentangling features and protected attributes. However, computing the mutual information for continuous variables is generally intractable (Paninski, 2003; Belghazi et al., 2018). Therefore, we follow Belghazi et al. (2018), and estimate the mutual information with an MLP $T_\psi$:

$$\mathcal{M}(\mathcal{D}; \psi, \theta) = \mathbb{E}_{(\mathbf{x}, g) \sim \mathcal{D}} T_\psi(\mathbf{z}, g) - \log \mathbb{E}_{(\mathbf{x}, \bar{g}) \sim \mathcal{D}} \left[ e^{T_\psi(\mathbf{z}, \bar{g})} \right], \tag{2}$$

where $T_\psi$ is a two-layer MLP with ReLU as non-linear activation, outputting a single scalar, $(\mathbf{x}, g) \sim \mathcal{D}$ refers to a sampling from the joint distribution, $(\mathbf{x}, \bar{g}) \sim \mathcal{D}$ refers to a sampling from the product of the marginal distributions. To sample from the product of the marginal distributions we sample twice from the joint distribution: $(\mathbf{x}, \bar{g}) \, \text{s.t.} \, (\mathbf{x}, g) \sim \mathcal{D}, (\bar{\mathbf{x}}, \bar{g}) \sim \mathcal{D}$, and keep only $\mathbf{x}$ and $\bar{g}$ as in Belghazi et al. (2018). We now use this definition to derive MIU (M̲utual I̲nformation-aware M̲achine U̲nlearning), our proposed method.

**Unlearning term.** We minimize the mutual information between forget set features and their group label to unlearn the forget set $\mathcal{D}_f$ while maintaining a good trade-off between information removal and robustness preservation. Therefore, we denote the unlearning term as:

$$\mathcal{M}(\mathcal{D}_f; \psi, \theta). \tag{3}$$

This term is high when the forget features correlate with the group information. Intuitively, we want this term to be low as this implies that we are unlearning the relation between forget-set features and labels while decorrelating the unlearning process from group information. Formally, minimizing this term achieves unlearning while preserving group robustness due to the following proposition:

**Proposition 1.** *Mutual information minimization is equivalent to cross-entropy loss maximization on a group classifier.*

*Proof.* Let us rewrite Eq. (1) in terms of entropy and conditional entropy:

$$I(Z; G) = \mathcal{H}(G) - \mathcal{H}(G|Z) \propto -\mathcal{H}(G|Z), \tag{4}$$

where group entropy $\mathcal{H}(G)$ is constant, thus, it can be ignored during optimization. Let $h_\xi^g(\mathbf{z}) = \hat{g}$ be a group classifier, namely a linear layer that classifies image groups using features $\mathbf{z}$, and let $\widehat{G}$ be the random variable associated with estimated groups. We can now write the cross-entropy loss for classifying groups using entropy notation as:

$$\mathcal{H}(G; \widehat{G}|Z) = \mathcal{H}(G|Z) + D_{\mathrm{KL}}(G\|\widehat{G}|Z). \tag{5}$$

Following Boudiaf et al. (2020), Lemma 2, we can relate Eqs. (4) and (5) by decoupling Eq. (5) into two steps, i.e., first optimize $D_{\mathrm{KL}}(G\|\widehat{G}|Z)$ and then $\mathcal{H}(G|Z)$. In other words, the first step fixes the backbone weights $\theta$ and optimizes the group classifier weights $\xi$, while the second only optimizes model weights $\theta$, keeping the classifier weights frozen. As the classifier is fixed in the latter, then maximizing the cross-entropy with respect to encoder weights corresponds to maximizing $\mathcal{H}(G|Z)$, i.e., minimizing $I(Z; G)$. Thus, minimizing $I(Z; G)$ (as in Eq. (3)) is equivalent to computing gradient ascent on group information, functionally disentangling label and spurious attributes from image features. $\square$

**Calibration term.** As unlearning might also affect other groups, we designed an extra term to improve group performance retention. Thus, we minimize the mutual information discrepancy between the original and unlearning model on the reweighted remaining dataset $\omega(\mathcal{D}_r)$:

$$\mathcal{L}_c(\omega(\mathcal{D}_r); \psi, \theta, \theta_o) = \|\mathcal{M}(\omega(\mathcal{D}_r); \psi, \theta) - \mathcal{M}(\omega(\mathcal{D}_r); \psi, \theta_o)\|^2, \tag{6}$$

where $\mathcal{M}(\omega(\mathcal{D}_r); \psi, \theta_o)$ estimates the mutual information using features computed by the original backbone $f_{\theta_o}(\cdot)$. Equation (6) encourages the unlearned model to mimic the original robustness, preserving its behavior across groups.

**Retaining term.** Since unlearning generally causes performance degradation (Kurmanji et al., 2024; Fan et al., 2024b; Jia et al., 2023), we perform gradient descent steps on the remaining data to ensure that the original model group accuracy is preserved after unlearning. Additionally, we use REWEIGHT to limit the degradation of group robustness. Therefore, we optimize the cross-entropy loss on the reweighted remaining dataset $\omega(\mathcal{D}_r)$:

$$\mathcal{L}_r(\omega(\mathcal{D}_r); \varphi, \theta) = \mathbb{E}_{(\mathbf{x},y)\sim\omega(\mathcal{D}_r)} \mathcal{L}_{\text{CE}}(\mathbf{x}, y; \varphi, \theta). \tag{7}$$

**Putting all together.** The *retaining term* in MIU preserves the model's original discriminative capabilities. The *unlearning term* removes $\mathcal{D}_f$ from model weights while minimizing performance loss for dominant groups in the forget set. Finally, the *calibration term* ensures that the unlearned model maintains its original robustness. Therefore:

$$\varphi_u, \theta_u = \arg\min_{\varphi,\theta} \underbrace{\mathcal{L}_r(\omega(\mathcal{D}_r); \varphi, \theta)}_{\text{Retaining term}} + \underbrace{\mathcal{M}(\mathcal{D}_f; \psi, \theta)}_{\text{Unlearning term}} + \lambda \cdot \underbrace{\mathcal{L}_c(\omega(\mathcal{D}_r); \psi, \theta, \theta_o)}_{\text{Calibration term}}. \tag{8}$$

**MIU pseudocode.** Algorithm 1 presents MIU pseudocode in a PyTorch-like (Paszke et al., 2019) style. We follow previous works (Kurmanji et al., 2024; Fan et al., 2024b) and compute *unlearning* and *retaining* steps separately. Thus, we alternate between computing an unlearning epoch using Eq. (3) and a retaining one using Eqs. (6) and (7). Like SCRUB (Kurmanji et al., 2024), we find it beneficial to stop performing unlearning steps after a predefined number of epochs (i.e., 5 out of 10). Contrary to Ragonesi et al. (2021), we do not update the mutual information representation at every step. Instead, we empirically observed that updating it via 100 gradient updates for the first epoch and 10 updates for the remaining iterations is sufficient to achieve satisfactory results and limit the required resources. As we keep the optimization steps fixed, the overall mutual information estimation overhead depends on the dataset size. For a small dataset like Waterbirds (Sagawa et al., 2020), we estimate an overhead of about $1.80\times$. Instead, for CelebA (Liu et al., 2015) and FairFace (Karkkainen & Joo, 2021), we estimate an increase of approximately $1.03\times$ in unlearning time, which is negligible.

---

**Algorithm 1** PyTorch-like MIU pseudocode.

```python
def MIU(model, mine, mine_original, optim, train_dataloader, remaining_dataloader, forget_dataloader):
    unlearned = copy.deepcopy(model)
    for epoch in range(epochs):
        # Update mine (mutual information neural estimator) as in Belghazi et al. (2018)
        tune_mine(unlearned, mine, train_dataloader)

        # Use mine to unlearn the forget_dataset for the first forget_epochs epochs
        if epoch < forget_epochs:
            for image, group in forget_dataloader:
                group_bar = torch.randint_like(group, num_groups)
                loss = mine(unlearned(image), group, group_bar)
                update_model(loss, optim)

        # Apply fine-tuning steps at every epoch using the reweighted sampler
        for image, target, group in reweight(remaining_dataloader):
            group_bar = torch.randint_like(group, num_groups)
            loss = F.cross_entropy(head(unlearned(image)), target)
            loss += lambda * F.mse_loss(
                mine(unlearned(image), group, group_bar),
                mine_original(model(image), group, group_bar).detach()
            )
            update_model(loss, optim)
    return unlearned
```

---

# 4 Experiments

This section describes the experimental protocol (Sec. 4.1) and compares MIU with multiple methods in group robust unlearning (Sec. 4.2). We then compare existing approaches and ours when varying the unlearning ratio of the forget set dominant group (Sec. 4.3) and when sampling the forget set from multiple groups (Sec. 4.4). Finally, Sec. 4.5 shows a complete ablation study of MIU's components.

## 4.1 Experimental protocol

**Datasets.** We follow established works in group-robust optimization (Sagawa et al., 2020; Idrissi et al., 2022; Liu et al., 2021; Park et al., 2022; Park & Byun, 2024) and conduct experiments on CelebA (Liu et al., 2015) and Waterbirds (Sagawa et al., 2020) datasets, setting *blonde* and *male* as the target and protected attributes for CelebA (Liu et al., 2015), while setting *waterbirds* and *land* as the target and sensitive attributes for Waterbirds (Sagawa et al., 2020). We additionally experiment with FairFace (Karkkainen & Joo, 2021), given its numerous annotations, using age as the downstream task and randomly choosing the class *20-29* and the ethnicity *afro-american* as target and protected attributes, respectively. Unless stated otherwise, forget-set images are sampled from groups described by the above target-sensitive attribute pairs. Moreover, unless stated otherwise, the forget set simulates the worst-case scenario where a single group is responsible for the unlearning request, leading to a high forget distribution imbalance. The unlearning ratio is defined as the proportion of samples from that particular group that have been unlearned. After unlearning, the model must have unlearned the forget data and maintained its original robustness.

**Hardware and training details.** We obtain PRETRAIN and RETRAIN by fine-tuning via empirical-risk minimization a ResNet-18 (He et al., 2016) pre-trained on ImageNet (Russakovsky et al., 2015) for 30 epochs, using SGD with 0.9 momentum and weight decay. The learning rate is decayed with a cosine annealing scheduler for the entire training. We additionally warm-up the learning rate for the first two epochs using a linear scheduler. We apply standard data augmentation techniques, namely, random resized crop, random horizontal flip, and input normalization (He et al., 2016). We limited fine-tuning to 10 epochs for approximate unlearning methods, searching for the optimal configuration for the other hyperparameters. The $\lambda$ parameter of MIU is set between 1 and 10 (see Appx. B.5). All experiments ran on a single A100 Nvidia GPU, using PyTorch (Paszke et al., 2019).

**Baselines.** We compare MIU against three state-of-the-art machine unlearning approaches. L1-SPARSE (Jia et al., 2023) forgets sensitive information by fine-tuning the original model on the remaining set with a sparsity regularization term. SALUN (Fan et al., 2024b) proposes a saliency-based unlearning that forgets data via random labeling. SCRUB (Kurmanji et al., 2024) minimizes the divergence between the unlearned and original model on the remaining set while maximizing it on the unlearning data. Following previous works (Kurmanji et al., 2024; Jia et al., 2023), we report PRETRAIN, and RETRAIN, which are computed by fine-tuning an ImageNet (Russakovsky et al., 2015) pre-trained ResNet-18 (He et al., 2016). We also report RETRAIN + GROUP-DRO (Sagawa et al., 2020) to validate the proposed RETRAIN + REWEIGHT.

We reimplemented all three baselines following the existing codebases. For L1-SPARSE (Jia et al., 2023), we perform 10 fine-tuning epochs on the remaining set with a linearly decaying $L1$ regularization that follows this rule: $\gamma_t = (1 - t/T)\gamma$, where $t$ is the epoch, $T$ is the total number of iterations, and $\gamma$ the initial penalty. For SALUN (Fan et al., 2024b) we followed the "small-scale" implementation of the original codebase (i.e., the implementation for CIFAR-10 (Krizhevsky et al., 2009) and SVHN (Netzer et al., 2011)) as it is meant for datasets with few classes. We first compute the saliency mask using the gradient information from the forget set, pruning 50% of the network weights. We tune the pruned weights for 10 epochs by alternating a full pass on the forget set and a full pass on the remaining set. Also, SCRUB (Kurmanji et al., 2024) is implemented by separating forget and retaining steps, following the original code. The forget step maximizes the KL divergence between the original and the unlearned model in the forget data set. Following the original paper, we stop computing it after 3, 5, or 7 epochs. Instead, the retaining step minimizes the linear combination between the cross-entropy loss and the KL divergence between the original and unlearned model, respectively scaled by 0.99 and 0.001, as reported in (Kurmanji et al., 2024). We note that all methods use the same dataset splits; therefore, they must unlearn the same forget set.

Table 1: **Group-robust machine unlearning in CelebA (Liu et al., 2015).** We build the forget set by sampling data points from a single group. The unlearning ratio is set to 0.5. We compare MIU against L1-SPARSE (Jia et al., 2023), SALUN (Fan et al., 2024b), and SCRUB (Kurmanji et al., 2024). The avg. gap and deltas are computed against RETRAIN + REWEIGHT, and we bold the methods that achieve the smallest discrepancy with it. Other reference models are in light gray.

| method | RW | RA | UA | TA | MIA | EO | GA | avg. gap ↑ |
|---|---|---|---|---|---|---|---|---|
| PRETRAIN | × | 96.2 | 41.9 | 95.9 | 0.9 | 24.2 | 40.6 | - |
| RETRAIN | × | 96.5 | 31.3 | 95.9 | 1.6 | 27.0 | 34.4 | - |
| RETRAIN + GDRO | × | 95.8 | 67.4 | 95.1 | 13.2 | 12.5 | 63.1 | - |
| RETRAIN | ✓ | 96.3 | 39.7 | 95.8 | 2.2 | 23.9 | 41.3 | - |
| L1-SPARSE (Jia et al., 2023) | × | 95.7 (0.5) | 29.0 (10.7) | 95.4 (0.3) | **1.5 (1.3)** | 28.5 (4.6) | 30.4 (10.9) | 95.3 |
| SALUN (Fan et al., 2024b) | × | **96.2 (0.2)** | 29.3 (10.3) | **95.8 (0.1)** | 0.7 (1.5) | 29.1 (5.2) | 30.6 (10.7) | 95.3 |
| SCRUB (Kurmanji et al., 2024) | × | 96.5 (0.3) | 35.1 (4.6) | 95.9 (0.2) | 0.6 (1.7) | 26.6 (2.8) | 35.9 (5.4) | 97.5 |
| MIU | × | 96.4 (0.3) | **36.3 (3.4)** | **95.9 (0.1)** | **1.0 (1.3)** | 26.1 (2.2) | 36.3 (5.0) | 98.0 |
| L1-SPARSE (Jia et al., 2023) | ✓ | 95.6 (0.6) | 37.3 (3.6) | 95.4 (0.4) | 0.7 (1.5) | 26.7 (2.8) | 34.8 (6.5) | 97.4 |
| SALUN (Fan et al., 2024b) | ✓ | **96.1 (0.2)** | 42.9 (8.2) | **95.8 (0.1)** | 0.6 (2.0) | 23.9 (3.4) | 41.1 (9.8) | 96.1 |
| SCRUB (Kurmanji et al., 2024) | ✓ | 96.4 (0.3) | 43.5 (3.8) | 96.0 (0.2) | 0.7 (1.5) | 23.7 (0.7) | 43.0 (1.7) | 98.6 |
| MIU | ✓ | 96.3 (0.3) | 43.2 (3.5) | 96.0 (0.2) | **1.2 (1.3)** | 24.0 (0.5) | 41.3 (0.4) | **99.0** |

**Metrics.** To evaluate unlearning and group-robustness, we rely on six different metrics. The first three are remaining (RA), forget (UA), and test (TA) accuracy. We also report the membership inference attack (MIA) (Yeom et al., 2018), which measures the MIA-Efficacy as described in (Jia et al., 2023; Fan et al., 2024b). Finally, we assess the change in group robustness by looking at equalized odds (EO) (Hardt et al., 2016), and the test accuracy of the forget-set dominant group (GA).

GA measures the ratio of correctly classified test samples that belong to the same dominant group(s) of the forget set. Therefore, let $\mathcal{D}_{te}^{g_f} = \{(\mathbf{x}_i, y_i, a_i) \mid g_f = g_i = (y_i, a_i) \wedge (\mathbf{x}_i, y_i, a_i) \in \mathcal{D}_{te}\}$ be the subset of the test set composed of all samples of group $g_f$, then GA is computed as follows:

$$\text{GA} = \frac{1}{|\mathcal{D}_{te}^{g_f}|} \sum_{i=1}^{|\mathcal{D}_{te}^{g_f}|} \mathbb{1}\left[(h_\varphi \circ f_\theta)(\mathbf{x}_i) = y_i\right], \tag{9}$$

where $\mathbb{1}$ is the indicator function, returning 1 if the argument is True, and 0 otherwise. Like other accuracy metrics, the closer GA is to 1 (or 100%), the better the model is at classifying data of the dominant group of the forget set.

Finally, to ease the interpretation of six metrics, we follow previous works (Jia et al., 2023; Fan et al., 2024b) and compute the average gap (**avg. gap**) and per-metric deltas with the gold standard.[3] Following them, we do not report whether metrics should be maximized or minimized as the machine unlearning objective is to reduce the discrepancy with the gold standard on each metric, except for avg. gap, which must be maximized. We use RETRAIN + REWEIGHT as the gold standard since it better reduces the gap with PRETRAIN in TA, EO, and GA, compared to RETRAIN and RETRAIN + GROUP-DRO (Sagawa et al., 2020). Further details are in Appx. A.

## 4.2 Results on group unlearning

Tables 1 to 3 show results for group-robust unlearning on CelebA (Liu et al., 2015), Waterbirds (Sagawa et al., 2020), and FairFace (Karkkainen & Joo, 2021) using an unlearning ratio $r$ of 0.5.

**CelebA.** RETRAIN + REWEIGHT achieves the best trade-off between original performance preservation and unlearning, showing the best gap with UA (-2.2), EO (-0.3), and GA (+0.7). The forget accuracy (UA) is very close to that of PRETRAIN, which might seem strange at first glance. Yet, UA and GA share similar values for both PRETRAIN and RETRAIN. This suggests that the model achieves a low generalization error, as the forget accuracy aligns with that of unseen samples (from the same group) *regardless* of whether the forget set was part of the training data. Additionally, as CelebA (Liu et al., 2015) counts numerous images,

---

[3]Deltas are computed per each seed and then averaged over three runs.

Table 2: **Group-robust machine unlearning in Waterbirds (Sagawa et al., 2020).** We build the forget set by sampling data points from a single group. The unlearning ratio is set to 0.5. We compare MIU against L1-SPARSE (Jia et al., 2023), SALUN (Fan et al., 2024b), and SCRUB (Kurmanji et al., 2024). The avg. gap and deltas are computed against RETRAIN + REWEIGHT, and we bold the methods that achieve the smallest discrepancy with it. To avoid confusion, other reference models are in light gray.

| method | RW | RA | UA | TA | MIA | EO | GA | avg. gap ↑ |
|---|---|---|---|---|---|---|---|---|
| PRETRAIN | × | 98.9 | 84.5 | 87.7 | 33.3 | 26.2 | 56.6 | - |
| RETRAIN | × | 98.7 | 52.4 | 86.5 | 54.8 | 30.4 | 49.4 | - |
| RETRAIN + GDRO | × | 94.7 | 89.3 | 91.6 | 21.4 | 7.3 | 83.1 | - |
| RETRAIN | ✓ | 99.0 | 59.5 | 87.2 | 53.6 | 28.3 | 51.6 | - |
| L1-SPARSE (Jia et al., 2023) | × | **99.0 (0.2)** | 59.5 (7.1) | 85.6 (1.6) | 44.0 (9.5) | 32.2 (4.1) | 48.8 (11.1) | 94.4 |
| SALUN (Fan et al., 2024b) | × | 100.0 (1.0) | 50.0 (9.5) | 81.8 (5.4) | 90.5 (36.9) | 38.7 (10.4) | 39.3 (12.3) | 87.4 |
| SCRUB (Kurmanji et al., 2024) | × | 98.8 (0.3) | 60.7 (10.7) | 86.9 (0.7) | 45.2 (10.7) | 31.9 (4.3) | 41.7 (9.8) | 93.9 |
| MIU | × | 100.0 (1.0) | 53.6 (8.3) | 86.1 (1.2) | 58.3 (7.1) | 28.3 (3.0) | 53.8 (7.5) | 95.3 |
| L1-SPARSE (Jia et al., 2023) | ✓ | **98.7 (0.2)** | 64.3 (11.9) | 85.0 (2.2) | 46.4 (7.1) | 30.6 (2.3) | 53.7 (8.3) | 94.7 |
| SALUN (Fan et al., 2024b) | ✓ | 100.0 (1.0) | 47.6 (16.7) | 81.1 (6.1) | 91.7 (38.1) | 39.0 (10.7) | 39.0 (12.5) | 85.8 |
| SCRUB (Kurmanji et al., 2024) | ✓ | **98.9 (0.2)** | 66.7 (11.9) | **87.0 (0.6)** | 44.0 (9.5) | 30.9 (3.4) | 44.3 (7.3) | 94.5 |
| MIU | ✓ | 99.9 (0.9) | **54.8 (4.8)** | 85.8 (1.4) | **59.5 (6.0)** | **28.3 (1.8)** | **53.7 (4.0)** | **96.9** |

Table 3: **Group-robust machine unlearning in FairFace (Karkkainen & Joo, 2021).** We build the forget set by sampling data points from a single group. The unlearning ratio is set to 0.5. We compare MIU against L1-SPARSE (Jia et al., 2023), SALUN (Fan et al., 2024b), and SCRUB (Kurmanji et al., 2024). The avg. gap and deltas are computed against RETRAIN + REWEIGHT, and we bold the methods that achieve the smallest discrepancy with it. To avoid confusion, other reference models are in light gray.

| method | RW | RA | UA | TA | MIA | EO | GA | avg. gap ↑ |
|---|---|---|---|---|---|---|---|---|
| PRETRAIN | × | 65.6 | 79.0 | 57.2 | 0.2 | 5.4 | 71.2 | - |
| RETRAIN | × | 66.8 | 57.8 | 56.5 | 0.9 | 9.2 | 58.7 | - |
| RETRAIN + GDRO | × | 61.7 | 56.3 | 51.4 | 10.2 | 2.3 | 57.4 | - |
| RETRAIN | ✓ | 66.7 | 69.3 | 56.7 | 0.7 | 5.6 | 69.6 | - |
| L1-SPARSE (Jia et al., 2023) | × | 64.0 (2.7) | 74.1 (4.8) | **56.9 (0.5)** | 0.2 (0.7) | 6.1 (0.9) | 69.4 (0.4) | 98.3 |
| SALUN (Fan et al., 2024b) | × | 66.3 (0.3) | 66.6 (3.0) | 55.9 (0.8) | 0.3 (0.6) | 9.0 (3.4) | 60.3 (9.3) | 97.1 |
| SCRUB (Kurmanji et al., 2024) | × | 66.9 (0.3) | 65.4 (3.9) | 56.7 (0.6) | 1.0 (0.5) | 9.9 (4.3) | 61.3 (8.3) | 97.0 |
| MIU | × | **66.7 (0.1)** | 74.7 (5.4) | 57.2 (0.8) | 0.3 (0.5) | 6.0 (1.1) | 66.1 (3.5) | 98.1 |
| L1-SPARSE (Jia et al., 2023) | ✓ | 64.4 (2.3) | 72.9 (3.6) | 56.0 (0.9) | 0.3 (0.4) | 6.1 (2.0) | 67.0 (7.1) | 97.3 |
| SALUN (Fan et al., 2024b) | ✓ | 65.1 (1.5) | 69.8 (5.6) | 54.8 (1.8) | 0.3 (0.4) | 6.6 (1.7) | 63.7 (5.9) | 97.2 |
| SCRUB (Kurmanji et al., 2024) | ✓ | 66.7 (0.2) | 73.4 (4.1) | 57.2 (0.6) | **0.7 (0.3)** | **6.2 (0.7)** | 70.2 (1.8) | **98.7** |
| MIU | ✓ | 64.7 (1.9) | **71.6 (2.3)** | **57.1 (0.5)** | **0.3 (0.3)** | 5.8 (1.5) | **70.3 (1.2)** | **98.7** |

REWEIGHT easily preserves original group accuracies, thus, recovering the original model performances and showing the same UA of PRETRAIN. Retraining the model with GROUP-DRO (Sagawa et al., 2020) generally leads to unwanted unlearning behaviors, i.e., the UA increases by 25.5 points. Plain RETRAIN, instead, suffers performance degradation in the dominant group of the forget set, as we highlight in Fig. 2 (GA lowers by 6.2 points). Adopting the proposed reweighting strategy leads to a better trade-off in group-robust unlearning.

MIU achieves the best performance preservation (Tab. 1) by scoring the highest GA alignment both when using ($\Delta 0.4$) and not using ($\Delta 5.0$) REWEIGHT. These results highlight that mutual information improves group performance retention. Instead, existing approaches must rely on REWEIGHT to close the gap in GA and EO with MIU as they are agnostic to group-robust machine unlearning. We also notice that using REWEIGHT increases the UA for all tested algorithms. Yet, despite the method used, coupling REWEIGHT with approximate unlearning always recovers most of the original GA. We visualize this in Fig. 4, which shows the same experiment as Fig. 2 while highlighting the REWEIGHT contribution. However, as MIU is strictly designed for this task, it generally achieves better robustness than existing approaches, scoring the best avg. gap.

**Waterbirds.** Similarly to CelebA (Liu et al., 2015) experiments, RETRAIN + REWEIGHT achieves the best discrepancy (Tab. 2) in terms of TA (-0.5), EO (+2.1), and GA (-5.0), better preserving original group robustness. RETRAIN + GROUP-DRO (Sagawa et al., 2020), instead, increases the UA by 4.8% above

PRETRAIN and by 36.9% above RETRAIN, which can negatively affect the unlearning evaluation if used as the gold standard for approximate machine unlearning. Compared to the CelebA (Liu et al., 2015) case, RETRAIN achieves a better unlearning-preservation trade-off. Nonetheless, RETRAIN + REWEIGHT is still a better candidate for group-robust unlearning, always achieving the best calibration with the original model.

In Waterbirds (Sagawa et al., 2020), our method outperforms existing methods both with and without REWEIGHT. Yet, REWEIGHT does not provide a substantial improvement as only the forget accuracy (from a delta of 8.3 to 4.8) and the EO (from a delta of 7.5 to 4.0) get significantly enhanced. Instead, REWEIGHT substantially increases L1-SPARSE (Jia et al., 2023) and SCRUB (Kurmanji et al., 2024) average UA (+4.8 and +6.0), achieving higher values than RETRAIN + REWEIGHT (59.5%). Nonetheless, it also improves the GA, showing better dominant forget group preservation. We argue that the subtle improvement provided by REWEIGHT is caused by the limited dataset size of Waterbirds (Sagawa et al., 2020). By increasing the sampling likelihood of the few dominant forget group images left, the network overfits those few samples, limiting robustness preservation. Even without REWEIGHT, MIU outperforms all baselines, regardless of whether they use REWEIGHT, further validating our design choices.

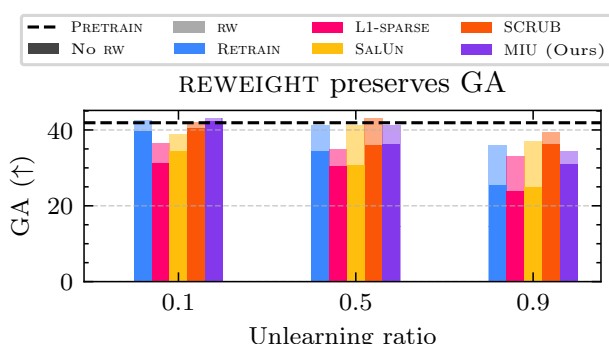

Figure 4: **REWEIGHT for group-robust unlearning.** As in Fig. 2, we test different methods and REWEIGHT in group-robust unlearning on CelebA. Darker colors are used for methods without the reweighting, while lighter ones correspond to methods coupled with REWEIGHT. As the unlearning ratio grows, the methods' GA degrade. Instead, adding REWEIGHT restores the original GA.

**FairFace.** Differently from the other datasets, here RETRAIN + GROUP-DRO (Sagawa et al., 2020) struggles to preserve original model accuracy (Tab. 3) in both the dominant forget group (-13.8) and the test set (-5.8). We ascribe this behavior to numerous FairFace (Karkkainen & Joo, 2021) groups that make the group-robust optimization challenging. Instead, RETRAIN + REWEIGHT overcomes this issue as it simply reweights group sampling probabilities to match the original training dataset statistics, achieving better GA (-1.6) and a TA (-0.5). Importantly, our experiments highlight a key advantage of REWEIGHT, which functions effectively "off the shelf" with the original model hyperparameters. Unlike RETRAIN + GROUP-DRO (Sagawa et al., 2020), it requires *no additional tuning*.

All methods show good alignment to RETRAIN + REWEIGHT, even without reweighting. TA and GA are generally preserved, with SALUN (Fan et al., 2024b) scoring lowest at 55.9% (delta 0.8) and 60.3% (delta 9.3) respectively. However, methods without REWEIGHT show higher UA than RETRAIN, indicating partial scrubbing of the forget set. Regardless, Sec. 4.5 shows that the high UA of MIU is not caused by poor unlearning but the *calibration term*, which recovers the original group robustness. REWEIGHT further enhances model robustness, reflected in higher GA and lower EO. Finally, MIU + REWEIGHT and SCRUB (Kurmanji et al., 2024) + REWEIGHT achieve the same avg. gap, but while our method shows a better UA and GA alignment with RETRAIN + REWEIGHT, SCRUB (Kurmanji et al., 2024) obtains a better alignment in EO terms. Although REWEIGHT does not fundamentally improve the avg. gap, it generally enhances GA and EO, promoting performance retention.

**Discussion.** UA strongly correlates with GA after unlearning in all three datasets. A high UA-GA correlation suggests that the forget set behaves as unseen data of the same group, indicating that the forget set was *properly* unlearned. Moreover, MIU consistently approximates RETRAIN+REWEIGHT better than existing methods or is on par, while REWEIGHT reliably preserves performance without drawbacks in model retraining or approximate unlearning.

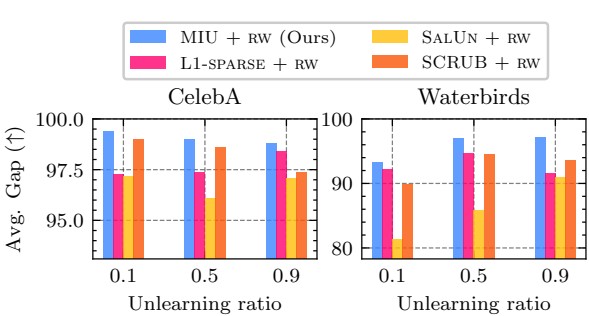
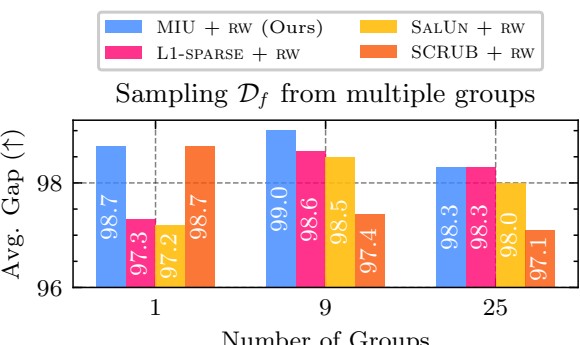

Figure 5: **Group-robust unlearning across different unlearning ratios.** We compare L1-SPARSE (Jia et al., 2023), SALUN (Fan et al., 2024b), and SCRUB (Kurmanji et al., 2024) against our approach while using the REWEIGHT strategy on all methods. MIU achieves overall the best avg. gap when varying the unlearning ratio.

Figure 6: **Sampling the forget set from multiple groups.** We evaluate our method against L1-SPARSE (Jia et al., 2023), SALUN (Fan et al., 2024b), and SCRUB (Kurmanji et al., 2024) when the forget set is sampled from multiple FairFace (Karkkainen & Joo, 2021) groups. MIU is more consistent across experiments, always achieving the best result.

### 4.3 Impact of different unlearning ratios

Figure 5 analyzes methods avg. gap across different unlearning ratios. On CelebA (Liu et al., 2015), all methods show consistent discrepancies from RETRAIN + REWEIGHT across different unlearning ratios, likely because REWEIGHT recovers a large portion of the lost robustness. Nonetheless, MIU consistently outperforms existing approaches, scoring 99.4%, 99.0%, and 98.8% at unlearning ratios of 0.1, 0.5, and 0.9. Instead, all algorithms struggle at 0.1 unlearning ratio in Waterbirds (Sagawa et al., 2020), where the small forget set size (i.e., five samples) causes high fluctuations in the UA, as each sample counts as $1/5$ of the total classification error. Furthermore, the reduced forget set size makes the gradient and BatchNorm (Ioffe, 2015) estimation noisy. As the unlearning ratio grows (i.e., 0.5 and 0.9), MIU outperforms baselines by a growing margin. At unlearning ratio 0.9, MIU achieves 97.2% on Waterbirds (vs. 93.6% of SCRUB (Kurmanji et al., 2024)). Furthermore, MIU remains more consistent across unlearning ratios, confirming that our design choices effectively narrow the avg. gap with RETRAIN + REWEIGHT. Full Fig. 5 tables are reported in Appx. B.

### 4.4 Multi-group unlearning

This section compares MIU and existing approaches in multi-group unlearning, i.e., when the forget set data is sampled from multiple groups (group composition is in Appx. B.3). Figure 6 shows the outcome of this experiment on FairFace (Karkkainen & Joo, 2021), given its numerous groups, with a varying number of groups in the forget set and an unlearning ratio fixed to 0.5. All methods achieve high scores and a good discrepancy with RETRAIN + REWEIGHT, though SCRUB (Kurmanji et al., 2024) performs worst (97.1% of avg. gap). Performance gaps shrink as more groups are included, as forget-set statistics align with the original training distribution, reducing non-uniform sampling effects. Nonetheless, MIU is the most consistent, highlighting its effectiveness in group-robust machine unlearning.

### 4.5 Ablations

Table 4 shows the ablation of MIU components in all three datasets by systematically adding each element to understand its contribution. We mark with "✓" when the component is used in the experiment and "×" when it is not. From left to right, we list *retaining term, unlearning term, calibration term,* and REWEIGHT. In the first row, we consider our method when

only the *unlearning term* and the *retaining term* are used. We highlight how the UA is low for all three datasets, demonstrating that mutual information minimization can be used to unlearn.

This baseline already achieves a remarkable avg. gap with RETRAIN + REWEIGHT, scoring a 97.5% in CelebA (Liu et al., 2015), which is the best among methods that do not use REWEIGHT. Adding our *calibration term* (Eq. (6)), leads to an increase in GA in all three datasets, with FairFace (Karkkainen & Joo, 2021) showing a growth of 6.9%. We highlight that the forget accuracy also grows. Yet, this increase is caused by Eq. (6), which calibrates the mutual information to match the original group robustness. Therefore, the high UA cannot be blamed on the poor unlearning. Although most previous approaches exploit a *retaining term* (Chundawat et al., 2023a; Kurmanji et al., 2024; Jia et al., 2023; Fan et al., 2024b), we also ablate this component for completeness. As previous works suggest (Kurmanji et al., 2024; Chundawat et al., 2023a),

Table 4: **MIU ablations.** We compute MIU ablations on each of the three investigated datasets. From left to right, we report the investigated dataset, *retaining term*, *unlearning term*, *calibration term*, and REWEIGHT. We measure performance using UA, GA, and avg. gap. The configuration that corresponds to MIU + REWEIGHT is highlighted.

| dataset | Eq. (7) | Eq. (3) | Eq. (6) | RW | UA | GA | gap ↑ |
|---|---|---|---|---|---|---|---|
| CelebA | ✓ | ✓ | ✗ | ✗ | 35.5 | 35.4 | 97.5 |
| | ✓ | ✓ | ✓ | ✗ | 36.3 | 36.3 | 98.0 |
| | ✗ | ✓ | ✓ | ✗ | 27.2 | 28.5 | 95.2 |
| | ✓ | ✓ | ✓ | ✓ | 43.2 | 41.3 | 99.0 |
| Waterbirds | ✓ | ✓ | ✗ | ✗ | 47.6 | 51.1 | 92.5 |
| | ✓ | ✓ | ✓ | ✗ | 53.6 | 53.8 | 95.3 |
| | ✗ | ✓ | ✓ | ✗ | 16.7 | 16.8 | 81.9 |
| | ✓ | ✓ | ✓ | ✓ | 54.8 | 53.7 | 96.9 |
| FairFace | ✓ | ✓ | ✗ | ✗ | 63.1 | 59.2 | 96.1 |
| | ✓ | ✓ | ✓ | ✗ | 74.7 | 66.1 | 98.1 |
| | ✗ | ✓ | ✓ | ✗ | 87.1 | 81.1 | 93.0 |
| | ✓ | ✓ | ✓ | ✓ | 71.6 | 70.3 | 98.7 |

removing the *retaining term* negatively impacts model utility, resulting in the lowest performance overall (e.g., 81.9% in Waterbirds). Similarly, when adding REWEIGHT, the avg. gap gets improved in all three datasets (e.g., 96.9% in Waterbirds), as we also highlight in Fig. 4. These results highlight the contribution of each of MIU components that allow for unlearning (Eq. (3)) while preserving forget set dominant group performance (Eq. (6)).

## 5 Conclusion

This paper is the first to address the performance degradation that can be caused by non-uniformly distributed forget sets in both model retraining and approximate unlearning. We show that adopting a simple data distribution reweighting (REWEIGHT) for retraining the model is a simple and better alternative than retraining with GROUP-DRO (Sagawa et al., 2020). Moreover, we propose the first approximate unlearning method (MIU) that unlearns personal data while reducing the risk of degradation of the forget set dominant group. Our evaluation demonstrates that RETRAIN + REWEIGHT consistently improves over a simple RETRAIN while MIU outperforms existing baselines in group-robust machine unlearning.

**Limitations.** One limitation of our work is the assumption that group annotations are known, which may not hold in real-world applications where such labels are difficult to obtain. Therefore, a natural follow-up of this work is a group-agnostic methodology that preserves model robustness as achieved by REWEIGHT. A naïve approach towards this direction would be to discover groups from data (Kim et al., 2024; D'Incà et al., 2024), and applying approaches presented in this paper. Furthermore, our evaluation is restricted to the classification setting. Applying the proposed techniques and baselines to other domains might be nontrivial, and there is no guarantee that unlearning effectiveness and accuracy will transfer directly. Despite these limitations, our work is an important first step toward understanding and mitigating the accuracy degradation caused by group-unbalanced forget sets.

**Broader impact statement.** Machine unlearning is designed to remove user data from a trained model. While its primary goal is to preserve privacy, it can also be misused for malicious purposes. Targeted unlearning of specific groups may lead to biased models with harmful consequences. However, the proposed *group-robust machine unlearning* seeks to minimize performance degradation for dominant groups in the forget set. Thus, MIU and REWEIGHT counteract biases introduced by unlearning, mitigating the negative societal impacts of its misuse.

**Acknowledgements.** We acknowledge the CINECA award under the ISCRA initiative for the availability of high-performance computing resources and support. Elisa Ricci and Massimiliano Mancini are supported by the MUR PNRR project FAIR - Future AI Research (PE00000013), funded by NextGeneration EU. Elisa Ricci is also supported by the EU projects AI4TRUST (No.101070190) and ELIAS (No.01120237). Thomas De Min is funded by NextGeneration EU. This work is also supported by the EU project ELLIOT (101214398) and by the French National Research Agency (ANR) with the ANR-20-CE23-0027. We thank Olivier Laurent for his valuable feedback and insightful suggestions.

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

# A Metrics

This section provides further details on the adopted metrics, i.e., the RA, UA, TA, the membership-inference attack (MIA) (Yeom et al., 2018), the dominant forget group accuracy (GA), the equalized odds (EO), and the average gap (avg. gap) (Fan et al., 2024b).

**RA, UA, and TA.** We evaluate the model on the remaining, forget, and test sets to compute the remaining, forget, and test accuracy. Therefore, we report the ratio of correctly classified samples for each of these subsets of the dataset.

**MIA.** To compute the membership inference attack, we follow previous works (Jia et al., 2023; Fan et al., 2024b) and train a model on remaining and validation losses to predict membership. Therefore, we assign a different binary label to remaining and validation losses and train the model to discriminate between them. After model training, we use such a model to infer the membership of forget set data. In all tables, we report the MIA-Efficacy (Jia et al., 2023) that is computed as follows:

$$\text{MIA-Efficacy} = \frac{TN}{|\mathcal{D}_f|}, \tag{10}$$

where $TN$ are the true negatives, i.e., the number of samples the MIA predicted as non-members. Instead of training a support vector machine as in (Jia et al., 2023; Fan et al., 2024b), we used a random forest as the accuracies are comparable with an SVM. Moreover, training is faster since it can be easily parallelized. The higher the MIA-Efficacy, the better the model's privacy protection.

**EO.** Equalized Odds (Hardt et al., 2016) measures model fairness or prediction dependencies on protected attributes. To compute the EO, we measure model performance discrepancy by varying the sensitive attribute value and averaging over different target labels. Formally, the EO for a binary classification model is computed as follows (Hardt et al., 2016):

$$\text{EO} = \frac{1}{2} \sum_{y=0}^{1} \left| P(\hat{Y} = 1 \mid Y = y, A = 0) - P(\hat{Y} = 1 \mid Y = y, A = 1) \right|, \tag{11}$$

where $\hat{Y}, Y, A$ are random variables describing model predictions, target attributes, and sensitive attributes. EO measures the absolute difference in outputting a positive prediction when the protected attribute equals 1 and 0, averaging over the two target attributes. In the FairFace (Karkkainen & Joo, 2021) case, we set all classes and protected attributes that do not match those of the dominant group of the forget set as $y = 0$ and $a = 0$, as FairFace counts more than two classes and more than two attributes. The lower the EO, the better the model fairness.

**avg. gap.** Following previous works (Jia et al., 2023; Fan et al., 2024b), we compute the avg. gap to simplify the comparison among different methodologies. avg. gap is computed as the average metric discrepancy between the unlearned model and the retrained gold standard. Formally, let $M = \{\text{RA}, \text{UA}, \text{TA}, \text{MIA}, \text{EO}, \text{GA}\}$ be the set of all metrics used in this paper, then avg. gap is computed as follows:

$$\text{avg. gap} = \overline{\sum}_{m \in M} 1 - \left| \overline{\sum}_{s \in S} m(\varphi_u^s, \theta_u^s) - m(\varphi_r^s, \theta_r^s) \right|, \tag{12}$$

where $m(\varphi_u^s, \theta_u^s)$ and $m(\varphi_r^s, \theta_r^s)$ are calculated using unlearned and retrained model weights, $s$ is the experiment seed and $\overline{\sum}$ is the average. The closer avg. gap is to 1 (or 100 in the tables), the better the approximation of the retrained model.

# B Additional results

This section contains the additional results that could not be included in the main paper. Appendix B.1 intuitively shows the accuracy degradation caused by non-uniformly sampled forget sets in all three investigated datasets. Appendices B.2 and B.3 report tables associated with experiments on different unlearning ratios and sampling the forget set from multiple groups. Appendix B.5 further expands MIU ablations. Finally, Appendix B.6 further evaluates MIU's and REWEIGHT's fairness and robustness preservation.

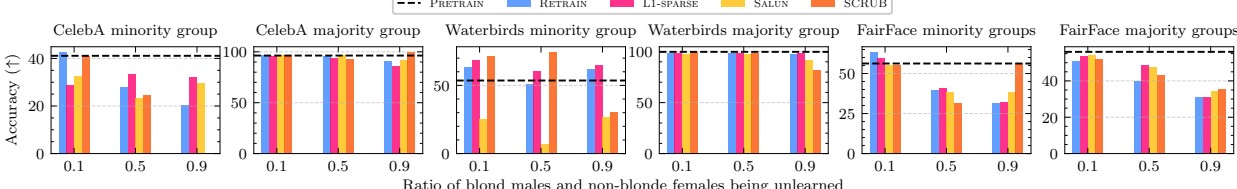

Figure 7: **Unlearning non-uniformly sampled data.** We test standard model retraining, and popular approximate unlearning methods (L1-SPARSE (Jia et al., 2023), SALUN (Fan et al., 2024b), SCRUB (Kurmanji et al., 2024)) in group-robust unlearning. The more samples of least represented groups are unlearned, the lower the model accuracy on such groups. On the contrary, the most represented ones are less affected.

Table 5: **Group-robust machine unlearning in CelebA (Liu et al., 2015) with 0.1 unlearning ratio.** We build the forget set by sampling data points from a single group. The unlearning ratio is set to 0.1. We compare MIU against L1-SPARSE (Jia et al., 2023), SALUN (Fan et al., 2024b), and SCRUB (Kurmanji et al., 2024). The avg. gap is computed against RETRAIN + REWEIGHT.

| method | RW | RA | UA | TA | MIA | EO | GA | avg. gap ↑ |
|---|---|---|---|---|---|---|---|---|
| PRETRAIN | × | 96.2±0.0 | 44.0±2.7 | 95.9±0.1 | 0.5±0.7 | 23.4±0.5 | 41.9±1.7 | - |
| RETRAIN | × | 96.2±0.0 | 39.9±1.0 | 95.8±0.0 | 0.2±0.3 | 24.6±0.3 | 39.6±0.7 | - |
| RETRAIN | ✓ | 96.2±0.0 | 44.4±2.1 | 95.9±0.0 | 0.7±0.6 | 23.4±0.7 | 42.4±2.2 | - |
| L1-SPARSE (Jia et al., 2023) | × | 95.4±0.1 | 36.0±3.9 | 95.4±0.1 | 0.7±0.0 | 27.6±0.7 | 31.3±4.4 | 95.7±1.2 |
| SALUN (Fan et al., 2024b) | × | 96.2±0.0 | 36.2±1.0 | 95.8±0.0 | 0.0±0.0 | 26.9±0.6 | 34.4±2.3 | 96.6±1.4 |
| SCRUB (Kurmanji et al., 2024) | × | 96.4±0.0 | 42.8±2.1 | 96.0±0.0 | 0.5±0.7 | 24.5±0.3 | 40.6±0.8 | 98.8±0.8 |
| MIU | × | 96.2±0.0 | 43.0±2.4 | 95.9±0.0 | 0.2±0.3 | 23.4±0.1 | 42.2±0.5 | 99.1±0.3 |
| L1-SPARSE (Jia et al., 2023) | ✓ | 95.4±0.0 | 38.9±1.9 | 95.4±0.0 | 1.0±0.3 | 26.3±1.4 | 36.5±4.1 | 97.3±1.0 |
| SALUN (Fan et al., 2024b) | ✓ | 96.1±0.0 | 40.1±5.9 | 95.9±0.0 | 0.7±0.6 | 25.0±1.9 | 38.7±6.1 | 97.2±0.9 |
| SCRUB (Kurmanji et al., 2024) | ✓ | 96.4±0.0 | 44.7±3.0 | 96.0±0.0 | 0.5±0.3 | 23.9±0.4 | 42.0±1.3 | 99.0±0.4 |
| MIU | ✓ | 96.2±0.0 | 44.9±2.7 | 95.9±0.0 | 0.2±0.3 | 22.9±0.5 | 43.0±0.7 | 99.4±0.1 |

## B.1 Unlearning non-uniformly sampled data

Figure 2 of the main paper shows how the forget set dominant group accuracy drops for least represented groups when increasing the unlearning ratio, limiting the analysis to CelebA (Liu et al., 2015) due to space constraints. This section reports the same experiment on all three investigated datasets for completeness.

Figure 7 shows accuracy variations when unlearning *blond males* (minority) and *non-blonde females* (majority) in CelebA (Liu et al., 2015), *waterbirds on land* (minority) and *landbirds on land* (majority) in Waterbirds (Sagawa et al., 2020), and *more than 70 y.o. Middle Easterns* (minority), *60-69 y.o. Caucasians* (minority), *20-29 y.o. African Americans* (majority), and *30-39 y.o. Southeast Asians* (majority) in FairFace (Karkkainen & Joo, 2021), with different unlearning ratios: 0.1, 0.5, 0.9. While the accuracy drop is more evident on CelebA (Liu et al., 2015), it is also visible in the other two datasets. Interestingly, some unlearning methods are more robust to this performance drop than others (e.g., SCRUB (Kurmanji et al., 2024) scores a 56.25% in FairFace (Karkkainen & Joo, 2021) with unlearning ratio 0.9 vs. 38.54% of SALUN (Fan et al., 2024b) in minority groups, where the latter is the second best).

Although some methods are more robust, we argue that the quality of the unlearning process influences the accuracy of the dominant group of the forget set. The less the forget set was unlearned, the more the performance retention. As an example, Sec. 4.2 highlights that likely SCRUB (Kurmanji et al., 2024) fails to effectively unlearn the forget set in the FairFace (Karkkainen & Joo, 2021) experiment, showing higher minority group accuracy (56.25%) than the RETRAIN (31.25%). Thus, to better investigate the relationship between unlearning effectiveness and performance retention, Appx. B.2 reports tables associated with Fig. 7 using all investigated metrics.

Table 6: **Group-robust machine unlearning in CelebA (Liu et al., 2015) with 0.5 unlearning ratio.** We build the forget set by sampling data points from a single group. The unlearning ratio is set to 0.5. We compare MIU against L1-SPARSE (Jia et al., 2023), SALUN (Fan et al., 2024b), and SCRUB (Kurmanji et al., 2024). The avg. gap is computed against RETRAIN + REWEIGHT.

| method | RW | RA | UA | TA | MIA | EO | GA | avg. gap ↑ |
|---|---|---|---|---|---|---|---|---|
| PRETRAIN | × | 96.2±0.2 | 41.9±0.8 | 95.9±0.1 | 0.9±0.3 | 24.2±1.0 | 40.6±2.1 | - |
| RETRAIN | × | 96.5±0.0 | 31.3±0.3 | 95.9±0.0 | 1.6±0.4 | 27.0±0.4 | 34.4±0.9 | - |
| RETRAIN | ✓ | 96.3±0.1 | 39.7±0.7 | 95.8±0.0 | 2.2±1.2 | 23.9±0.8 | 41.3±1.6 | - |
| L1-SPARSE (Jia et al., 2023) | × | 95.7±0.1 | 29.0±3.1 | 95.4±0.1 | 1.5±0.6 | 28.5±0.6 | 30.4±2.9 | 95.3±0.7 |
| SALUN (Fan et al., 2024b) | × | 96.2±0.1 | 29.3±7.7 | 95.8±0.1 | 0.7±0.2 | 29.1±1.5 | 30.6±7.5 | 95.3±2.5 |
| SCRUB (Kurmanji et al., 2024) | × | 96.5±0.2 | 35.1±1.7 | 95.9±0.0 | 0.6±0.2 | 26.6±1.1 | 35.9±2.0 | 97.5±0.4 |
| MIU | × | 96.4±0.1 | 36.3±0.9 | 95.9±0.1 | 1.0±0.3 | 26.1±0.8 | 36.3±2.0 | 98.0±0.4 |
| L1-SPARSE (Jia et al., 2023) | ✓ | 95.6±0.0 | 37.3±3.8 | 95.4±0.1 | 0.7±0.3 | 26.7±0.9 | 34.8±4.8 | 97.4±1.4 |
| SALUN (Fan et al., 2024b) | ✓ | 96.1±0.1 | 42.9±10.5 | 95.8±0.0 | 0.6±0.5 | 23.9±2.8 | 41.1±9.8 | 96.1±2.0 |
| SCRUB (Kurmanji et al., 2024) | ✓ | 96.4±0.2 | 43.5±0.1 | 96.0±0.1 | 0.7±0.1 | 23.7±0.7 | 43.0±0.9 | 98.6±0.3 |
| MIU | ✓ | 96.3±0.2 | 43.2±0.6 | 96.0±0.0 | 1.2±0.1 | 24.0±0.5 | 41.3±1.1 | 99.0±0.2 |

Table 7: **Group-robust machine unlearning in CelebA (Liu et al., 2015) with 0.9 unlearning ratio.** We build the forget set by sampling data points from a single group. The unlearning ratio is set to 0.9. We compare MIU against L1-SPARSE (Jia et al., 2023), SALUN (Fan et al., 2024b), and SCRUB (Kurmanji et al., 2024). The avg. gap is computed against RETRAIN + REWEIGHT.

| method | RW | RA | UA | TA | MIA | EO | GA | avg. gap ↑ |
|---|---|---|---|---|---|---|---|---|
| PRETRAIN | × | 96.4±0.1 | 44.0±6.6 | 95.9±0.1 | 0.7±0.1 | 23.5±1.3 | 41.1±5.2 | - |
| RETRAIN | × | 96.7±0.0 | 20.2±1.0 | 95.9±0.0 | 7.2±1.4 | 31.5±0.6 | 25.4±1.9 | - |
| RETRAIN | ✓ | 96.4±0.1 | 33.6±1.1 | 95.7±0.1 | 4.3±1.3 | 25.6±0.7 | 35.9±1.9 | - |
| L1-SPARSE (Jia et al., 2023) | × | 95.9±0.1 | 21.6±6.6 | 95.4±0.1 | 3.9±1.0 | 31.5±1.6 | 23.9±7.1 | 94.7±2.6 |
| SALUN (Fan et al., 2024b) | × | 96.6±0.1 | 20.8±0.8 | 95.9±0.0 | 2.5±0.8 | 31.5±0.6 | 25.0±2.4 | 94.6±0.8 |
| SCRUB (Kurmanji et al., 2024) | × | 96.7±0.1 | 27.2±1.2 | 95.9±0.0 | 3.1±0.5 | 31.3±0.4 | 26.3±0.7 | 96.0±0.6 |
| MIU | × | 96.5±0.1 | 32.6±0.9 | 95.8±0.1 | 1.5±0.2 | 27.9±0.3 | 30.9±1.1 | 98.1±0.7 |
| L1-SPARSE (Jia et al., 2023) | ✓ | 95.9±0.0 | 33.2±2.9 | 95.2±0.1 | 2.6±0.7 | 28.3±0.5 | 33.1±1.8 | 98.4±0.6 |
| SALUN (Fan et al., 2024b) | ✓ | 96.4±0.1 | 38.1±6.7 | 95.8±0.2 | 1.9±1.2 | 25.9±1.1 | 36.9±7.2 | 97.1±0.4 |
| SCRUB (Kurmanji et al., 2024) | ✓ | 96.7±0.1 | 41.9±1.5 | 95.9±0.1 | 1.7±0.1 | 24.9±0.2 | 39.3±0.9 | 97.4±0.2 |
| MIU | ✓ | 96.7±0.1 | 35.2±2.0 | 95.8±0.0 | 4.2±1.1 | 26.9±1.4 | 34.3±2.7 | 98.8±0.4 |

## B.2 Impact of different unlearning ratios

For completeness purposes, this section reports all tables associated with Figs. 2, 4, 5 and 7. For these experiments, the main paper summarizes the results to reduce the occupied space and simplify the interpretation (i.e., limiting the reported metrics to one). Therefore, Tabs. 5, 7, 8, 10, 11 and 13 present experiments in CelebA (Liu et al., 2015), Waterbirds (Sagawa et al., 2020), and FairFace (Karkkainen & Joo, 2021) datasets, with unlearning ratios of 0.1, and 0.9. Additionally, Tabs. 6, 9 and 12 report the standard deviations of Tabs. 1 to 3.

Overall, we notice that in CelebA (Liu et al., 2015), the higher the unlearning ratio, the lower the forget accuracy (from 43.0% to 32.6% using MIU), with the gap being reduced when REWEIGHT is included in the retaining step (from 44.9% to 35.2% with MIU). RA, TA, and MIA remain stable across different unlearning ratios. Instead, EO behaves similarly to GA and UA, with high values at high unlearning ratios (e.g., MIU scores 22.9% vs. 26.9% with 0.1 and 0.9 unlearning ratios).

In Waterbirds (Sagawa et al., 2020), metrics do not show global trends, except for EO and GA, which worsen as the unlearning ratio grows. Therefore, methods lose accuracy on the dominant group of the forget set as the ratio of unlearned samples grows, with SCRUB (Kurmanji et al., 2024) showing the highest drop (from 44.3% to 25.1%). Similarly, GA and EO worsen even when adding REWEIGHT. As the unlearning ratio grows, the number of forget set dominant group samples left in the remaining set lowers. Given the small size of Waterbirds (Sagawa et al., 2020) (4795 samples, of which 56 are in the smallest group), REWEIGHT strongly increases the sampling likelihood of a restricted number of samples to preserve original accuracy, causing overfitting. Thus, the overall benefit of REWEIGHT is reduced (GA increases by only 1.3 with RETRAIN).

Table 8: **Group-robust machine unlearning in Waterbirds (Sagawa et al., 2020) with 0.1 unlearning ratio.** We build the forget set by sampling data points from a single group. The unlearning ratio is set to 0.1. We compare MIU against L1-sparse (Jia et al., 2023), SalUn (Fan et al., 2024b), and SCRUB (Kurmanji et al., 2024). The avg. gap is computed against Retrain + reweight.

| method | RW | RA | UA | TA | MIA | EO | GA | avg. gap ↑ |
|---|---|---|---|---|---|---|---|---|
| Pretrain | × | 99.0±0.1 | 73.3±18.9 | 88.0±0.5 | 53.3±9.4 | 25.9±0.7 | 57.6±2.0 | - |
| Retrain | × | 99.0±0.2 | 46.7±24.9 | 87.1±0.8 | 60.0±28.3 | 27.4±1.7 | 57.4±2.9 | - |
| Retrain | ✓ | 98.7±0.3 | 46.7±24.9 | 87.3±0.7 | 66.7±24.9 | 26.7±1.3 | 57.6±4.8 | - |
| L1-sparse (Jia et al., 2023) | × | 99.0±0.1 | 60.0±16.3 | 85.3±0.8 | 46.7±24.9 | 29.2±1.7 | 57.9±2.6 | 92.3±3.0 |
| SalUn (Fan et al., 2024b) | × | 100.0±0.0 | 53.3±9.4 | 78.3±3.3 | 66.7±9.4 | 42.7±5.1 | 33.3±7.3 | 81.6±6.7 |
| SCRUB (Kurmanji et al., 2024) | × | 98.9±0.1 | 53.3±9.4 | 86.9±0.4 | 53.3±18.9 | 31.1±1.1 | 44.3±3.5 | 91.3±2.0 |
| MIU | × | 99.1±0.0 | 60.0±16.3 | 87.1±0.2 | 33.3±18.9 | 26.1±2.1 | 59.2±5.3 | 91.2±9.1 |
| L1-sparse (Jia et al., 2023) | ✓ | 99.0±0.1 | 73.3±18.9 | 85.3±1.2 | 53.3±24.9 | 29.1±2.8 | 58.2±3.8 | 92.2±3.0 |
| SalUn (Fan et al., 2024b) | ✓ | 99.9±0.1 | 53.3±9.4 | 76.7±2.9 | 86.7±9.4 | 45.9±4.7 | 29.6±7.4 | 81.3±2.9 |
| SCRUB (Kurmanji et al., 2024) | ✓ | 98.8±0.2 | 53.3±9.4 | 86.8±0.7 | 60.0±16.3 | 31.5±0.8 | 42.8±3.5 | 89.8±0.4 |
| MIU | ✓ | 100.0±0.0 | 73.3±18.9 | 87.3±0.3 | 73.3±24.9 | 26.2±0.3 | 61.7±0.8 | 93.3±0.7 |

Table 9: **Group-robust machine unlearning in Waterbirds (Sagawa et al., 2020) with 0.5 unlearning ratio.** We build the forget set by sampling data points from a single group. The unlearning ratio is set to 0.5. We compare MIU against L1-sparse (Jia et al., 2023), SalUn (Fan et al., 2024b), and SCRUB (Kurmanji et al., 2024). The avg. gap is computed against Retrain + reweight.

| method | RW | RA | UA | TA | MIA | EO | GA | avg. gap ↑ |
|---|---|---|---|---|---|---|---|---|
| Pretrain | × | 98.9±0.3 | 84.5±1.7 | 87.7±0.5 | 33.3±6.1 | 26.2±1.9 | 56.6±6.0 | - |
| Retrain | × | 98.7±0.3 | 52.4±8.9 | 86.5±0.2 | 54.8±9.4 | 30.4±0.5 | 49.4±1.6 | - |
| Retrain | ✓ | 99.0±0.1 | 59.5±11.8 | 87.2±0.3 | 53.6±8.7 | 28.3±2.0 | 51.6±6.0 | - |
| L1-sparse (Jia et al., 2023) | × | 99.0±0.1 | 59.5±8.9 | 85.6±0.4 | 44.0±11.8 | 32.2±1.8 | 48.8±7.4 | 94.4±0.3 |
| SalUn (Fan et al., 2024b) | × | 100.0±0.0 | 50.0±5.1 | 81.8±0.4 | 90.5±3.4 | 38.7±1.2 | 39.3±3.3 | 87.4±3.5 |
| SCRUB (Kurmanji et al., 2024) | × | 98.8±0.2 | 60.7±7.7 | 86.9±0.6 | 45.2±8.9 | 31.9±1.8 | 41.7±1.7 | 93.9±1.4 |
| MIU | × | 100.0±0.0 | 53.6±7.7 | 86.1±1.0 | 58.3±8.9 | 28.3±1.7 | 53.8±2.6 | 95.3±0.7 |
| L1-sparse (Jia et al., 2023) | ✓ | 98.7±0.1 | 64.3±5.8 | 85.0±1.2 | 46.4±12.7 | 30.6±1.4 | 53.7±4.3 | 94.7±1.1 |
| SalUn (Fan et al., 2024b) | ✓ | 100.0±0.0 | 47.6±4.5 | 81.1±1.9 | 91.7±7.3 | 39.0±2.2 | 39.0±1.6 | 85.8±4.2 |
| SCRUB (Kurmanji et al., 2024) | ✓ | 98.9±0.2 | 66.7±1.7 | 87.0±0.5 | 44.0±8.9 | 30.9±1.3 | 44.3±3.1 | 94.5±1.3 |
| MIU | ✓ | 99.9±0.1 | 54.8±14.7 | 85.8±0.7 | 59.5±12.1 | 28.3±2.9 | 53.7±3.8 | 96.9±1.6 |

Also FairFace (Karkkainen & Joo, 2021) shows a general drop in UA, which grows when reweight is applied. EO and GA also behave like in CelebA (Liu et al., 2015), with an enhanced degradation at higher unlearning ratios. However, MIU shows good robustness even without reweight, scoring EO and GA that are close to Retrain+reweight, e.g., 10.4% vs. 11.9% in EO and 56.8% vs. 53.9 in GA with a 0.9 unlearning ratio (Tab. 13).

### B.3 Multi-group unlearning

This section further expands the experimental protocol of Sec. 4.4, highlighting the sampling composition and reporting all investigated metrics. For the 9 groups experiment, we sampled the forget set data with either *20-29 y.o., 50-59 y.o.*, and *3-9 y.o.* target attributes, and *Afro-American, Latino-Hispanic*, and *Caucasian* sensitive attributes. Instead, for the 25 groups experiment, we additionally sample from the *more than 70 y.o.* and *30-39 y.o.* target attributes, and *East Asian* and *Middle Eastern* sensitive attributes. The choice of groups is random, and the unlearning ratio is fixed to 0.5.

Full results for unlearning multiple groups in group-robust unlearning are reported in Tabs. 14 and 15. We notice overall the same trend as in Fig. 6. reweight contribution gets less important as the number of groups from which the forget set is sampled decreases, highlighted by unchanged UA, EO, and GA. The GA, when using MIU, e.g., increases by nearly 5% in the 9 groups experiment (Tab. 14), while it remains unchanged in the 25 groups one (Tab. 15). Test accuracies are better preserved when unlearning 1 or 9 groups, while we notice a drop (about 1.5%) in the 25 groups experiment. We argue that this decline is

Table 10: **Group-robust machine unlearning in Waterbirds (Sagawa et al., 2020) with 0.9 un-learning ratio.** We build the forget set by sampling data points from a single group. The unlearning ratio is set to 0.9. We compare MIU against L1-SPARSE (Jia et al., 2023), SALUN (Fan et al., 2024b), and SCRUB (Kurmanji et al., 2024). The avg. gap is computed against RETRAIN + REWEIGHT.

| method | RW | RA | UA | TA | MIA | EO | GA | avg. gap ↑ |
|---|---|---|---|---|---|---|---|---|
| PRETRAIN | × | 98.6±0.6 | 76.0±9.1 | 86.5±0.4 | 44.7±4.1 | 28.3±1.9 | 55.9±5.2 | - |
| RETRAIN | × | 98.9±0.2 | 41.3±5.7 | 84.3±0.3 | 68.7±6.8 | 36.4±1.4 | 41.7±3.9 | - |
| RETRAIN | ✓ | 98.9±0.1 | 41.3±2.5 | 85.7±0.2 | 62.7±3.4 | 33.5±1.3 | 43.0±2.9 | - |
| L1-SPARSE (Jia et al., 2023) | × | 98.9±0.2 | 60.0±3.3 | 82.9±1.2 | 50.7±5.0 | 35.3±0.4 | 49.9±4.6 | 92.9±1.0 |
| SALUN (Fan et al., 2024b) | × | 100.0±0.0 | 40.0±4.3 | 81.3±0.9 | 92.7±3.4 | 41.4±1.3 | 30.8±2.2 | 89.6±2.0 |
| SCRUB (Kurmanji et al., 2024) | × | 97.8±0.1 | 30.7±1.9 | 86.1±0.5 | 52.7±3.4 | 36.6±1.0 | 25.1±1.5 | 92.7±0.7 |
| MIU | × | 100.0±0.0 | 66.7±5.7 | 85.7±0.7 | 58.7±4.7 | 32.1±1.5 | 49.8±3.3 | 93.4±1.3 |
| L1-SPARSE (Jia et al., 2023) | ✓ | 99.0±0.2 | 59.3±10.6 | 84.6±0.6 | 45.3±6.8 | 31.5±0.7 | 55.0±4.0 | 91.5±4.1 |
| SALUN (Fan et al., 2024b) | ✓ | 100.0±0.0 | 45.3±2.5 | 80.3±0.7 | 87.3±1.9 | 41.8±0.5 | 31.9±4.8 | 90.9±1.0 |
| SCRUB (Kurmanji et al., 2024) | ✓ | 98.0±0.1 | 33.3±3.4 | 86.2±0.7 | 54.7±5.2 | 35.9±1.5 | 28.0±3.4 | 93.6±1.2 |
| MIU | ✓ | 98.9±0.2 | 44.7±3.4 | 83.1±1.3 | 65.3±3.4 | 35.7±2.2 | 45.0±1.7 | 97.2±0.3 |

Table 11: **Group-robust machine unlearning in FairFace (Karkkainen & Joo, 2021) with 0.1 unlearning ratio.** We build the forget set by sampling data points from a single group. The unlearning ratio is set to 0.1. We compare MIU against L1-SPARSE (Jia et al., 2023), SALUN (Fan et al., 2024b), and SCRUB (Kurmanji et al., 2024). The avg. gap is computed against RETRAIN + REWEIGHT.

| method | RW | RA | UA | TA | MIA | EO | GA | avg. gap ↑ |
|---|---|---|---|---|---|---|---|---|
| PRETRAIN | × | 66.2±0.7 | 79.2±2.5 | 57.2±0.1 | 0.7±0.2 | 5.8±0.2 | 71.6±2.1 | - |
| RETRAIN | × | 67.3±0.1 | 71.7±0.8 | 57.0±0.4 | 1.0±0.8 | 5.4±1.1 | 69.0±2.8 | - |
| RETRAIN | ✓ | 66.8±0.1 | 72.0±1.7 | 56.8±0.4 | 0.9±0.5 | 4.3±0.6 | 71.1±0.6 | - |
| L1-SPARSE (Jia et al., 2023) | × | 63.7±0.3 | 78.9±3.5 | 56.1±0.8 | 0.0±0.0 | 5.5±2.6 | 69.7±2.2 | 97.3±0.7 |
| SALUN (Fan et al., 2024b) | × | 65.9±0.8 | 73.9±3.9 | 55.1±1.1 | 0.5±0.0 | 2.9±1.1 | 69.8±7.0 | 97.8±0.9 |
| SCRUB (Kurmanji et al., 2024) | × | 68.4±0.5 | 78.7±0.8 | 57.5±0.3 | 0.2±0.2 | 5.7±0.2 | 70.4±1.5 | 97.9±0.5 |
| MIU | × | 66.9±0.5 | 81.3±0.2 | 57.3±0.2 | 0.2±0.2 | 5.3±0.6 | 70.4±0.5 | 97.8±0.5 |
| L1-SPARSE (Jia et al., 2023) | ✓ | 64.0±0.3 | 72.7±0.7 | 56.4±0.6 | 0.2±0.2 | 5.3±1.2 | 69.1±0.7 | 98.6±0.3 |
| SALUN (Fan et al., 2024b) | ✓ | 66.2±0.4 | 80.1±1.5 | 55.3±0.4 | 0.2±0.2 | 4.7±0.9 | 73.3±4.6 | 97.1±0.3 |
| SCRUB (Kurmanji et al., 2024) | ✓ | 68.4±0.5 | 79.2±1.0 | 57.5±0.4 | 0.2±0.2 | 5.6±1.1 | 70.9±1.7 | 97.8±0.6 |
| MIU | ✓ | 67.4±0.5 | 82.3±1.3 | 57.6±0.3 | 0.0±0.0 | 6.0±0.7 | 71.2±0.5 | 97.6±0.8 |

caused by the larger forget set, which reduces the number of available samples in the remaining set. Overall, MIU approximates RETRAIN + REWEIGHT better than baselines, consistently achieving the best avg. gap.

### B.4 Unlearning on MultiNLI

Table 16 shows the result for group-robust unlearning on MultiNLI (Williams et al., 2018) using an unlearning ratio $r$ of 0.5. REWEIGHT allows a successful recovery of the original group accuracy (-0.5), while preserving the test accuracy (-0.6). Compared to PRETRAIN, the EO are worse (14.4 vs. 16.2) but better compared to plain RETRAIN (16.2 vs.19.3), thus, REWEIGHT partly recovered also the equalized odds. MIU achieves the best performance preservation overall by scoring the highest GA alignment, both when using (96.0) and not using (95.8) REWEIGHT. Particularly, MIU is less influenced by REWEIGHT, and most of the alignment comes from its unique loss formulation. SCRUB (Kurmanji et al., 2024) achieves the second-best result without REWEIGHT (95.3), but struggles when used (92.5), as forget accuracy unexpectedly grows to high values (82.1). Instead, SALUN (Fan et al., 2024b) scores the best avg. gap after MIU (95.9), but shows high variance and low results without REWEIGHT. Overall, MIU is the most consistent among the two configurations and generally shows the lowest variance across metrics.

### B.5 Extended ablation study

Section 4.5 shows a comprehensive ablation of MIU's components. However, Tab. 4 limits the analysis only to the UA, GA, and avg. gap to reduce space usage. Thus, Tab. 17 reports all metrics investigated for completeness. Although Tab. 4 metrics are limited, we chose a subset that shows great variance along

Table 12: **Group-robust machine unlearning in FairFace (Karkkainen & Joo, 2021) with 0.5 unlearning ratio.** We build the forget set by sampling data points from a single group. The unlearning ratio is set to 0.5. We compare MIU against L1-sparse (Jia et al., 2023), SalUn (Fan et al., 2024b), and SCRUB (Kurmanji et al., 2024). The avg. gap is computed against Retrain + reweight.

| method | RW | RA | UA | TA | MIA | EO | GA | avg. gap ↑ |
|---|---|---|---|---|---|---|---|---|
| Pretrain | ✗ | 65.6±0.7 | 79.0±1.2 | 57.2±0.4 | 0.2±0.1 | 5.4±1.7 | 71.2±2.4 | - |
| Retrain | ✗ | 66.8±0.4 | 57.8±3.3 | 56.5±0.1 | 0.9±0.2 | 9.2±0.9 | 58.7±3.0 | - |
| Retrain | ✓ | 66.7±0.2 | 69.3±0.5 | 56.7±0.2 | 0.7±0.5 | 5.6±1.5 | 69.6±0.7 | - |
| L1-sparse (Jia et al., 2023) | ✗ | 64.0±0.3 | 74.1±1.2 | 56.9±0.5 | 0.2±0.1 | 6.1±0.7 | 69.4±0.7 | 98.3±0.2 |
| SalUn (Fan et al., 2024b) | ✗ | 66.3±0.4 | 66.6±3.4 | 55.9±0.6 | 0.3±0.1 | 9.0±0.5 | 60.3±2.4 | 97.1±1.1 |
| SCRUB (Kurmanji et al., 2024) | ✗ | 66.9±0.1 | 65.4±1.6 | 56.7±0.7 | 1.0±0.0 | 9.9±1.3 | 61.3±2.5 | 97.0±0.5 |
| MIU | ✗ | 66.7±0.2 | 74.7±1.2 | 57.2±0.7 | 0.3±0.0 | 6.0±2.0 | 66.1±4.4 | 98.1±0.4 |
| L1-sparse (Jia et al., 2023) | ✓ | 64.4±0.1 | 72.9±2.1 | 56.0±0.9 | 0.3±0.1 | 6.1±2.1 | 67.0±6.8 | 97.3±0.3 |
| SalUn (Fan et al., 2024b) | ✓ | 65.1±0.4 | 69.8±6.3 | 54.8±0.6 | 0.3±0.2 | 6.6±2.1 | 63.7±3.4 | 97.2±0.4 |
| SCRUB (Kurmanji et al., 2024) | ✓ | 66.7±0.1 | 73.4±2.2 | 57.2±0.5 | 0.7±0.3 | 6.2±1.1 | 70.2±2.7 | 98.7±0.7 |
| MIU | ✓ | 64.7±0.3 | 71.6±2.8 | 57.1±0.3 | 0.3±0.2 | 5.8±0.4 | 70.3±1.6 | 98.7±0.8 |

Table 13: **Group-robust machine unlearning in FairFace (Karkkainen & Joo, 2021) with 0.9 unlearning ratio.** We build the forget set by sampling data points from a single group. The unlearning ratio is set to 0.9. We compare MIU against L1-sparse (Jia et al., 2023), SalUn (Fan et al., 2024b), and SCRUB (Kurmanji et al., 2024). The avg. gap is computed against Retrain + reweight.

| method | RW | RA | UA | TA | MIA | EO | GA | avg. gap ↑ |
|---|---|---|---|---|---|---|---|---|
| Pretrain | ✗ | 66.0±0.0 | 77.5±2.1 | 56.6±0.4 | 0.2±0.0 | 5.5±1.1 | 69.0±2.8 | - |
| Retrain | ✗ | 67.3±0.6 | 38.5±1.8 | 56.0±0.4 | 2.7±0.3 | 23.1±1.5 | 37.1±2.1 | - |
| Retrain | ✓ | 67.1±0.4 | 53.6±1.2 | 56.6±0.4 | 1.8±0.1 | 11.9±1.0 | 53.9±0.7 | - |
| L1-sparse (Jia et al., 2023) | ✗ | 64.5±0.2 | 57.1±2.1 | 55.2±0.6 | 0.4±0.1 | 13.0±0.7 | 51.0±1.9 | 97.7±0.3 |
| SalUn (Fan et al., 2024b) | ✗ | 65.7±0.5 | 46.5±6.2 | 53.9±0.1 | 0.5±0.1 | 15.3±1.0 | 42.8±4.9 | 95.2±1.8 |
| SCRUB (Kurmanji et al., 2024) | ✗ | 60.2±1.0 | 52.7±4.4 | 53.3±0.5 | 2.4±0.7 | 15.6±1.4 | 48.7±4.9 | 95.7±0.6 |
| MIU | ✗ | 68.2±0.3 | 64.4±2.5 | 56.5±0.5 | 0.5±0.1 | 10.4±0.7 | 56.8±2.6 | 97.0±1.0 |
| L1-sparse (Jia et al., 2023) | ✓ | 64.0±0.4 | 74.5±3.0 | 55.9±0.4 | 0.3±0.2 | 5.6±0.6 | 69.8±4.6 | 91.9±1.5 |
| SalUn (Fan et al., 2024b) | ✓ | 65.5±0.5 | 66.1±3.7 | 55.3±0.2 | 0.5±0.3 | 7.2±1.2 | 62.8±4.9 | 94.9±1.7 |
| SCRUB (Kurmanji et al., 2024) | ✓ | 61.2±1.1 | 65.5±3.5 | 54.5±0.3 | 1.3±0.1 | 9.5±0.9 | 64.4±2.7 | 94.5±1.6 |
| MIU | ✓ | 64.7±0.2 | 67.1±1.4 | 56.7±0.2 | 0.5±0.1 | 8.8±0.2 | 63.5±1.6 | 94.9±0.6 |

Table 14: **Group-robust machine unlearning in FairFace (Karkkainen & Joo, 2021) by sampling from 9 groups.** We build the forget set by sampling data points from 9 groups. The unlearning ratio is set to 0.5. We compare MIU against L1-sparse (Jia et al., 2023), SalUn (Fan et al., 2024b), and SCRUB (Kurmanji et al., 2024). The avg. gap is computed against Retrain + reweight.

| method | RW | RA | UA | TA | MIA | EO | GA | avg. gap ↑ |
|---|---|---|---|---|---|---|---|---|
| Pretrain | ✗ | 64.6±0.5 | 81.5±0.5 | 57.4±0.3 | 0.4±0.0 | 1.1±0.1 | 72.5±0.7 | - |
| Retrain | ✗ | 66.5±0.5 | 60.1±0.9 | 55.4±0.4 | 1.1±0.1 | 5.6±0.4 | 60.4±1.7 | - |
| Retrain | ✓ | 64.4±1.1 | 72.1±2.2 | 56.3±0.3 | 0.5±0.0 | 2.0±0.6 | 71.8±2.7 | - |
| L1-sparse (Jia et al., 2023) | ✗ | 63.5±0.6 | 69.9±3.9 | 55.3±0.1 | 0.5±0.0 | 2.6±1.0 | 64.4±4.4 | 97.7±1.1 |
| SalUn (Fan et al., 2024b) | ✗ | 64.3±0.5 | 64.6±1.2 | 54.0±0.1 | 0.2±0.1 | 3.6±0.3 | 59.7±1.5 | 96.0±0.6 |
| SCRUB (Kurmanji et al., 2024) | ✗ | 67.2±0.4 | 74.3±0.7 | 56.9±0.2 | 0.3±0.1 | 1.8±0.7 | 65.6±0.6 | 97.7±0.3 |
| MIU | ✗ | 66.3±0.4 | 74.2±0.4 | 56.8±0.5 | 0.3±0.1 | 1.7±0.4 | 65.7±0.6 | 97.9±0.4 |
| L1-sparse (Jia et al., 2023) | ✓ | 63.7±0.1 | 75.2±0.6 | 56.3±0.1 | 0.2±0.1 | 1.3±0.3 | 69.6±1.2 | 98.6±0.2 |
| SalUn (Fan et al., 2024b) | ✓ | 63.7±0.8 | 74.4±1.5 | 55.5±0.4 | 0.4±0.0 | 2.3±0.3 | 69.0±1.6 | 98.4±0.2 |
| SCRUB (Kurmanji et al., 2024) | ✓ | 66.7±0.4 | 80.5±0.1 | 57.4±0.5 | 0.4±0.1 | 1.7±0.3 | 71.3±0.4 | 97.4±0.3 |
| MIU | ✓ | 63.4±0.4 | 73.2±0.3 | 56.7±0.3 | 0.4±0.1 | 1.5±0.5 | 70.3±1.0 | 99.0±0.1 |

Table 15: **Group-robust machine unlearning in FairFace (Karkkainen & Joo, 2021) by sampling from 25 groups.** We build the forget set by sampling data points from 25 groups. The unlearning ratio is set to 0.5. We compare MIU against L1-sparse (Jia et al., 2023), SaLUn (Fan et al., 2024b), and SCRUB (Kurmanji et al., 2024). The avg. gap is computed against Retrain + reweight.

| method | RW | RA | UA | TA | MIA | EO | GA | avg. gap ↑ |
|---|---|---|---|---|---|---|---|---|
| Pretrain | × | 65.1±0.3 | 70.6±0.5 | 56.9±0.3 | 0.3±0.1 | 1.6±0.6 | 62.2±1.4 | - |
| Retrain | × | 66.5±1.0 | 55.5±2.1 | 54.8±0.7 | 0.8±0.1 | 1.9±0.3 | 56.0±2.1 | - |
| Retrain | ✓ | 66.1±0.6 | 60.5±0.7 | 55.5±0.4 | 0.7±0.1 | 2.3±0.3 | 60.5±0.5 | - |
| L1-sparse (Jia et al., 2023) | × | 65.5±0.8 | 61.8±0.3 | 55.5±0.5 | 0.5±0.1 | 1.6±0.8 | 57.0±0.9 | 98.7±0.2 |
| SaLUn (Fan et al., 2024b) | × | 64.5±0.5 | 60.0±3.8 | 55.2±1.0 | 0.5±0.1 | 1.1±0.4 | 57.0±3.8 | 98.0±0.7 |
| SCRUB (Kurmanji et al., 2024) | × | 66.7±0.6 | 62.4±1.0 | 55.4±0.3 | 0.3±0.1 | 1.7±0.9 | 55.2±0.6 | 98.4±0.2 |
| MIU | × | 66.8±0.2 | 64.8±0.5 | 56.4±0.5 | 0.3±0.1 | 1.7±0.7 | 57.5±1.1 | 98.3±0.2 |
| L1-sparse (Jia et al., 2023) | ✓ | 64.3±0.7 | 66.7±0.4 | 55.8±0.4 | 0.4±0.1 | 2.1±0.8 | 60.8±1.0 | 98.3±0.3 |
| SaLUn (Fan et al., 2024b) | ✓ | 62.5±1.6 | 64.6±2.0 | 54.6±0.8 | 0.3±0.1 | 2.2±0.4 | 60.1±2.0 | 98.0±0.3 |
| SCRUB (Kurmanji et al., 2024) | ✓ | 65.3±0.3 | 71.4±0.6 | 57.0±0.3 | 0.2±0.0 | 1.9±0.9 | 63.7±0.7 | 97.1±0.1 |
| MIU | ✓ | 66.2±1.6 | 63.0±1.9 | 54.3±0.6 | 0.9±0.3 | 3.5±0.9 | 57.6±3.3 | 98.3±0.1 |

Table 16: **Group-robust machine unlearning in MultiNLI (Williams et al., 2018) with 0.5 unlearning ratio.** We build the forget set by sampling data points from a single group. The unlearning ratio is set to 0.5. We compare MIU against L1-sparse (Jia et al., 2023), SaLUn (Fan et al., 2024b), and SCRUB (Kurmanji et al., 2024). The avg. gap is computed against Retrain + reweight.

| method | RW | RA | UA | TA | MIA | EO | GA | avg. gap ↑ |
|---|---|---|---|---|---|---|---|---|
| Pretrain | × | 94.7±2.5 | 81.8±8.4 | 82.3±0.1 | 31.4±0.9 | 14.4±1.8 | 59.3±2.7 | - |
| Retrain | × | 95.9±1.1 | 53.0±3.7 | 82.3±0.1 | 47.6±3.9 | 19.3±0.6 | 50.8±3.1 | - |
| Retrain | ✓ | 95.3±1.1 | 59.8±0.7 | 81.7±0.0 | 43.3±2.3 | 16.2±0.3 | 58.8±2.0 | - |
| L1-sparse (Jia et al., 2023) | × | 84.8±2.0 | 46.6±4.8 | 79.0±0.7 | 34.1±2.3 | 23.3±1.4 | 42.0±2.1 | 90.1±1.8 |
| SaLUn (Fan et al., 2024b) | × | 76.4±30.3 | 36.5±25.9 | 63.8±21.4 | 36.1±25.6 | 20.2±0.8 | 35.1±24.9 | 80.8±19.0 |
| SCRUB (Kurmanji et al., 2024) | × | 96.4±1.9 | 56.8±4.6 | 82.4±0.2 | 46.7±5.5 | 22.6±1.7 | 47.6±3.3 | 95.3±1.3 |
| MIU | × | 99.2±0.1 | 60.3±1.6 | 79.2±0.2 | 54.9±2.8 | 18.8±0.7 | 55.6±2.0 | 95.8±0.3 |
| L1-sparse (Jia et al., 2023) | ✓ | 84.9±2.0 | 55.4±3.8 | 79.0±0.6 | 31.7±1.4 | 19.3±1.1 | 49.2±1.6 | 93.0±1.2 |
| SaLUn (Fan et al., 2024b) | ✓ | 96.0±0.1 | 54.3±3.6 | 78.3±0.2 | 48.1±2.0 | 18.5±1.9 | 51.9±2.6 | 95.9±2.3 |
| SCRUB (Kurmanji et al., 2024) | ✓ | 96.5±1.9 | 82.1±7.8 | 82.4±0.2 | 32.2±8.9 | 14.0±3.8 | 62.4±6.3 | 92.5±2.8 |
| MIU | ✓ | 99.9±0.0 | 67.0±2.6 | 81.5±0.2 | 51.1±2.6 | 18.0±0.4 | 56.6±1.2 | 96.0±0.8 |

different components and is more interesting to evaluate. For instance, EO variations are more nuanced compared to GA (e.g., it drops by 4.5 in FairFace (Karkkainen & Joo, 2021), while GA grows by 11.1).

The MIA shows subtle oscillations in CelebA (Liu et al., 2015) and FairFace (Karkkainen & Joo, 2021) while it drops when adding the *calibration term* in Waterbirds (Sagawa et al., 2020), getting closer to Retrain + reweight value (59.5% of MIU vs. 53.6% of Retrain + reweight). Finally, TA and RA are relatively stable across components, with RA showing a negative trend (64.7% using all three components vs. 65.6% of Pretrain).

Fig. 8, instead, reports the ablation study on parameter $\lambda$. We show results for $\lambda \in \{0, 1, 5, 10\}$ and average each experiment over three different seeds. Overall, all three datasets benefit from the calibration term. However, while CelebA (Liu et al., 2015) and FairFace (Karkkainen & Joo, 2021) achieve better results when $\lambda = 1$ (i.e., an avg. gap of 99.2 and 98.7), Waterbirds (Sagawa et al., 2020) minimizes the gap when $\lambda = 10$ (i.e., an avg. gap of 96.9). Nonetheless, results are generally stable, and when in doubt, we suggest setting $\lambda = 1$ as it always shows an alignment improvement with Retrain + reweight.

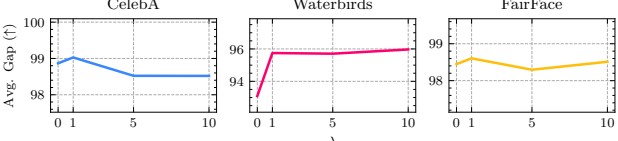

Figure 8: **Ablating parameter $\lambda$.** MIU avg. gap when varying parameter $\lambda$ in CelebA (Liu et al., 2015), Waterbirds (Sagawa et al., 2020), and FairFace (Karkkainen & Joo, 2021). While $\lambda = 1$ is optimal in CelebA (Liu et al., 2015) and FairFace (Karkkainen & Joo, 2021), Waterbirds (Sagawa et al., 2020) benefits from higher lambdas.

Table 17: **MIU ablations.** We compute MIU ablations on each of the three investigated datasets. From left to right, we report the investigated dataset, the *retaining term*, the *unlearning term*, the *calibration term*, and REWEIGHT. We measure performance using all metrics. The configuration that corresponds to MIU + REWEIGHT is highlighted.

| dataset | Eq. (7) | Eq. (3) | Eq. (6) | RW | RA | UA | TA | MIA | EO | GA | avg. gap ↑ |
|---|---|---|---|---|---|---|---|---|---|---|---|
| CelebA | ✓ | ✓ | × | × | 96.4±0.1 | 35.5±0.9 | 95.9±0.0 | 1.1±0.1 | 26.7±0.9 | 35.4±2.3 | 97.5±0.5 |
| | ✓ | ✓ | ✓ | × | 96.4±0.1 | 36.3±0.9 | 95.9±0.1 | 1.0±0.3 | 26.1±0.8 | 36.3±2.0 | 98.0±0.4 |
| | × | ✓ | ✓ | × | 95.8±0.5 | 27.8±5.7 | 95.3±0.4 | 0.9±0.2 | 25.5±0.8 | 28.5±6.9 | 95.2±2.5 |
| | ✓ | ✓ | ✓ | ✓ | 96.3±0.2 | 43.2±0.6 | 96.0±0.0 | 1.2±0.1 | 24.0±0.5 | 41.3±1.1 | 99.0±0.2 |
| Waterbirds | ✓ | ✓ | × | × | 100.0±0.0 | 47.6±7.3 | 85.0±0.6 | 73.8±3.4 | 32.3±0.8 | 51.1±0.6 | 92.5±4.6 |
| | ✓ | ✓ | ✓ | × | 100.0±0.0 | 53.6±7.7 | 86.1±1.0 | 58.3±8.9 | 28.3±1.7 | 53.8±2.6 | 95.3±0.7 |
| | × | ✓ | ✓ | × | 93.0±3.3 | 16.7±9.4 | 80.3±2.9 | 64.3±12.7 | 35.8±7.8 | 16.8±8.9 | 81.9±5.5 |
| | ✓ | ✓ | ✓ | ✓ | 99.9±0.1 | 54.8±14.7 | 85.8±0.7 | 59.5±12.1 | 28.3±2.9 | 53.7±3.8 | 96.9±1.6 |
| FairFace | ✓ | ✓ | × | × | 65.2±0.1 | 63.1±1.6 | 56.9±0.3 | 0.3±0.0 | 10.3±0.8 | 59.2±1.9 | 96.1±0.8 |
| | ✓ | ✓ | ✓ | × | 66.7±0.2 | 74.7±1.2 | 57.2±0.7 | 0.3±0.0 | 6.0±2.0 | 66.1±4.4 | 98.1±0.4 |
| | × | ✓ | ✓ | × | 59.1±3.1 | 87.1±6.8 | 54.5±1.8 | 0.0±0.0 | 3.1±0.5 | 81.1±6.2 | 93.0±3.1 |
| | ✓ | ✓ | ✓ | ✓ | 64.7±0.3 | 71.6±2.8 | 57.1±0.3 | 0.3±0.2 | 5.8±0.4 | 70.3±1.6 | 98.7±0.8 |

Table 18: **Additional Fairness Metrics.** Fairness metrics are computed on each of the three investigated datasets (using the same splits as Tabs. 1 to 3). From left to right, we report the method, DP, EP, EO, and WG. MIU + REWEIGHT is highlighted.

| method | DP | EP | EO | WG |
|---|---|---|---|---|
| **CelebA (Liu et al., 2015)** | | | | |
| PRETRAIN | 18.7 | 45.3 | 24.2 | 40.6 |
| RETRAIN | 18.9 (0.1) | 51.0 (5.7) | 27.0 (2.9) | 34.4 (-6.1) |
| RETRAIN+RW | 18.8 (0.1) | 44.6 (-0.7) | 23.9 (-0.3) | 41.3 (0.7) |
| MIU | 18.8 (0.1) | 45.0 (-0.3) | 24.0 (-0.1) | 41.3 (0.7) |
| **Waterbirds (Sagawa et al., 2020)** | | | | |
| PRETRAIN | 20.6 | 36.1 | 26.1 | 56.6 |
| RETRAIN | 23.2 (2.6) | 43.4 (7.3) | 30.4 (4.3) | 49.4 (-7.2) |
| RETRAIN + RW | 21.5 (0.9) | 40.6 (4.5) | 28.3 (2.2) | 51.6 (-5.0) |
| MIU | 22.9 (2.3) | 38.1 (2.0) | 28.3 (2.2) | 53.7 (-2.9) |
| **FairFace (Karkkainen & Joo, 2021)** | | | | |
| PRETRAIN | 2.0 | 7.6 | 5.4 | 9.4 |
| RETRAIN | 5.3 (3.3) | 17.9 (10.3) | 9.2 (3.8) | 8.3 (-1.1) |
| RETRAIN + RW | 3.8 (1.8) | 7.6 (0.0) | 5.6 (0.2) | 6.1 (-3.3) |
| MIU | 1.1 (-0.9) | 8.0 (0.4) | 5.8 (0.4) | 16.3 (6.9) |

## B.6 Additional Fairness Metrics

To further study the method's fairness after unlearning. We investigate three additional fairness metrics alongside the Equalized Odds (EO) (Hardt et al., 2016), namely, Demographic Parity (Kusner et al., 2017), Equal Opportunity (Hardt et al., 2016), and Worst Group Accuracy (WG) (Sagawa et al., 2020; Liu et al., 2021). To satisfy the Demographic Parity, the model's probability of outputting a positive prediction must be independent of the sensitive attribute. Therefore, we measure it as: $DP = |P(\hat{Y} = 1 \mid A = 0) - P(\hat{Y} = 1 \mid A = 1)|$. Similarly, to satisfy Equal Opportunity (EP), the model true positive rate must be independent of the sensitive attribute: $EP = |P(\hat{Y} = 1 \mid Y = 1, A = 0) - P(\hat{Y} = 1 \mid Y = 1, A = 1)|$. Instead, the Worst Group Accuracy measures the average accuracy of the worst group of the test set. Thus, the Worst Group Accuracy is computed as: $WG = \min_{g_i} \left\{ \frac{1}{|\mathcal{D}_{te}^{g_i}|} \sum_{i=1}^{|\mathcal{D}_{te}^{g_i}|} \mathbb{1}\left[(h_\varphi \circ f_\theta)(\mathbf{x}_i) = y_i\right] \right\}$, where $\mathcal{D}_{te}^{g_i}$ is the subset of images of group $g_i$ of the test set. The lower the first two metrics (DP and EP), the better the model fairness, while the higher the WG, the higher the model robustness.

Table 18 reports our evaluation using the additional metrics on CelebA (Liu et al., 2015), Waterbirds (Sagawa et al., 2020), and FairFace (Karkkainen & Joo, 2021). While plain model retraining generally shows performance degradation on almost all fairness metrics (e.g., +5.7 EP, +2.9 EO, and -6.1 WG, in CelebA (Liu et al., 2015)), using REWEIGHT recovers the original model fairness and overall robustness (i.e., -0.7 EP, -0.3 EO, and +0.7 WG, in CelebA (Liu et al., 2015)). Similarly, the proposed MIU preserves PRETRAIN

robustness by approximating RETRAIN + REWEIGHT (i.e., -0.3 EP, -0.1 EO, and +0.7 WG, in CelebA (Liu et al., 2015)).

