# OpenReview forum: "Group-robust Machine Unlearning"
_TMLR — Accepted by TMLR_

### Review · Reviewer_xfyW · 2025-08-07

**Summary Of Contributions:**

This paper studies the intersection of machine unlearning and group robustness, namely when the forget set consists primarily of data with a particular sensitive attribute. Benchmarks are provided which show that typical unlearning methods disproportionately harm group accuracy. Two methods are proposed to rectify this: in the exact unlearning setting, a sample reweighting scheme is shown to be sufficient, whereas in the approximate unlearning setting, a mutual-information-based algorithm called MIU is proposed. Experiments and ablations are then provided across three vision datasets.

**Summary of strengths:**
1. The problem is well-motivated by recent scholarship in AI ethics.
2. This paper has potential to bring two distinct research communities closer together.
3. The proposed MIU method is clearly explained, and the analysis and ablation studies are well-done.

**Summary of weaknesses:**
1. The benchmarking study in Section 3.2/App. B.1 should be expanded.
2. The claim of an iid forget set assumption in the literature is insufficiently supported.
3. The attractiveness classification task is ethically concerning.

**Additional Comments:**

1. The color-coding of the target and protected attributes is a nice touch but somewhat inconsistent, e.g., it is missing from the appendix.
2. I found the heavy use of both the terms “retain” and “retrain” confusing. I often had to re-read sentences to confirm the context, and it generally interrupted the flow of the writing. I would recommend changing one of these terms to something more distinct.
3. Typo in Section 3.3: “reweighing” should be “reweighting”?
4. In Algorithm 1, it would be nice to explain some acronyms in the caption. I assumed “dl” stands for “dataloader” and “mine” stands for mutual information neural estimation.
5. I recommend removing or rewording the following sentence: “The popularity of group-DRO (Sagawa et al., 2020) in group-robust optimization suggests it should minimize drops in the forget-set dominant group accuracy.” There is no justification for the popularity of a method being correlated with its performance on a particular task.
6. The bolding in the tables doesn’t make much sense, as the bolded numbers are often neither the lowest or highest in each column. It would be helpful to describe what is bolded in the caption and keep it consistent throughout the paper (bolding is also missing in the appendix tables). Relatedly, it would be helpful to include up-arrows and down-arrows in the tables to signify whether each metric is preferably large or small.
7. Section 3.4 is very well-written; I appreciate the step-by-step motivation of each term and inclusion of theoretical justification when appropriate. The pseudocode is also helpful.

**Audience:**

Yes

**Audience Explanation:**

A strength of this paper is its potential to bring two distinct research communities closer together, namely in machine unlearning and group robustness. Each community would likely be interested in this paper: the machine unlearning community for its interesting challenge and proposal of robustness-preserving unlearning algorithms, and the group robustness community for the application of debiasing techniques in a new application area. There is potential for future work in this direction, especially towards the question at the end of Section 3.3 -- “Was the knowledge unlearned if forget-set accuracy increased?” -- which may motivate sample-by-sample metrics of unlearning and examination of the interaction between unlearning and Group DRO.

**Broader Impact Concerns:**

No broader impact statement is provided.

I have an ethical concern regarding the primary application on the CelebA dataset, namely the use of machine learning to classify facial attractiveness. Attractiveness classification is a fundamentally problematic and outdated task which reinforces culturally biased and highly subjective standards of beauty. It risks amplifying social inequities with respect to gender, age, race, etc and does not align with modern standards of responsible AI development -- especially for a paper which focuses on group fairness. Moreover, substantial scholarship exists regarding the biased nature of the “attractive” attribute in CelebA specifically, e.g., [1, 2]. I strongly encourage the authors to reconsider the inclusion of this task; one alternative is to keep CelebA but switch the target attribute to hair color (which is more commonly used in the spurious correlations literature).

[1] Prabhu et al. “Covering up bias with Markov blankets: A post-hoc cure for attribute prior avoidance”. ICML 2019 Workshop on Invertible Neural Networks and Normalizing Flows.

[2] Rajabi et al. “Through a fair looking-glass: mitigating bias in image datasets.” HCII 2023.

**Claims And Evidence:**

No

**Claims Explanation:**

For the most part, the claims are well-supported, especially the analysis and ablation studies in Section 4. There are two claims which need more evidence, detailed below.

**Retain Set Group Distribution Harms Group Accuracy**

The following claims in Section 3.2 are insufficiently supported: “The more samples of a group are removed, the lower the resulting group accuracy. Intuitively, removing non-uniformly distributed data changes the retain set group distribution, ultimately harming the model’s generalization performance on the dominant group of the forget set.”

First, the supporting experiments in Figure 7 consist of learning only a single group in each of the three datasets, and each of these are minority groups/underrepresented in the training dataset (to my knowledge). It would be helpful to repeat these experiments for more groups (say, 4) in each dataset, distributed among both majority and minority groups.

Second, the claim “the more samples of a group are removed, the lower the resulting group accuracy” may not hold in situations where samples of *multiple groups* are being unlearned (c.f. the results in Section 4.4/Figure 6). Consider the following example: dataset D has a majority group A with 900 samples and a minority group B with 100 samples. If 100 samples from group A are unlearned, the experiments suggest the accuracy on group A will decrease. But if 200 samples from group A are unlearned and 100 samples from group B are unlearned, I would hypothesize it is possible for the accuracy on group A to *increase*, as the retain set would comprise solely group A.

Third, it is unclear whether the distribution of data across the groups in the retain set is the primary causation of the harm to generalization on the dominant group of the forget set (as opposed to, for example, less data overall in the retain set). To support this claim, I would expect controlled experiments varying the retain set group distribution while keeping confounding factors constant, with, e.g., correlation coefficients between group balance metrics and GA. Since this claim is not critical to the narrative, I may advise reducing the claim for this third point.

**IID Forget Set Assumption**

It is unclear to what extent the iid forget set assumption is actually prevalent in the machine unlearning literature, and it would be helpful to clearly delineate which methods specifically rely on this assumption. To my understanding, it seems that many of the cited methods in Section 2 actually allow for *arbitrary* forget sets, rather than requiring forget sets to be drawn randomly over the dataset. For example, SCRUB [1] studies two different settings, one where an entire class is forgotten and one where iid samples from a class are forgotten, and neither of these require the forget set to be sampled iid across all groups or the entire dataset.

With the tenuous nature of the iid forget set assumption in mind, I would argue that the primary contribution of this paper is actually its revision of the machine unlearning objective function. While previous unlearning methods prioritize metrics such as average accuracy on the retained data and average generalization [1], this work proposes to use group accuracy instead. Indeed, the calibration term of MIU explicitly enforces preservation of the base-model group accuracies, particularly on the worst-group and forget-set group.

Overall, I would encourage the authors to clarify their primary contribution, specify the extent of the iid forget set assumption, and consider repositioning the work as a revision of the machine unlearning accuracy objective.

[1] Kurmanji et al. “Towards Unbounded Machine Unlearning.” NeurIPS 2023.

**Requested Changes:**

**Critical for acceptance**
1. Expand the benchmarking study (see Claims).
2. Clarify the primary contribution of the paper with respect to the iid forget set assumption (see Claims).
3. Reconsider the attractiveness classification task (see Broader Impact Concerns).

**Would strengthen the work:**
1. Formally define the “unlearning ratio”. I assumed that it is the proportion of samples from a particular group which are to be unlearned (e.g., the forget set is 100% waterbirds on land, which comprises 90% of waterbirds on land in the train set), but it could also be interpreted as the proportion of samples from a particular group in the forget set (e.g., the forget set is 90% waterbirds on land and 10% other groups).
2. Idrissi et al. [1] extensively discuss a version of REWEIGHT, called by them RWG, but this paper only mentions their subsampling method in Section 2. It would be helpful to contrast REWEIGHT with RWG and discuss how this paper’s insights compare to [1]. In general, contextualization with the literature on reweighting methods would be helpful given REWEIGHT’s massive importance in obtaining good performance (c.f. Figure 4).
3. It took me a couple re-reads of the paper to understand that the most desired metric, which MIU is designed to maximize, is the *dominant forget-group accuracy (GA)*. The presentation is somewhat muddled by the discussion around Group DRO (which actually targets WGA) and the 5 other metrics in the tables. I would recommend bringing the definition of GA (Eq. 11) out of the appendix into Section 3 and increasing clarity around targeted metrics.
4. Various improvements to the writing (see Additional Comments).

[1] Idrissi et al. “Simple data balancing achieves competitive worst-group-accuracy.” CLeaR 2022.

---

> ### Author Response · Authors · 2025-09-23
>
> ### About how the retain set group distribution harms accuracy
> 1. We thank the reviewer for suggesting this experiment. Figures 2 and 7 of the revised paper report the outcome of the suggested evaluation, where we respectively unlearned *blond males* (minority group) and *non-blonde females* (majority group) in CelebA, *waterbirds on land* (minority) and *landbirds on land* (majority) in Waterbirds, and more than 70 y.o. Middle Easterns (minority), 60-69 y.o. Caucasians (minority), 20-29 y.o. African Americans (majority), and 30-39 y.o. Southeast Asians (majority) in FairFace, at varying unlearning ratios. As the first two datasets count four groups, we unlearned only two, while we unlearned four groups in FairFace. In our experiment, data from the majority groups are less affected by unlearning, as their ratio remains relatively high in the retain set, which can reduce model biases [1]. On the contrary, the accuracy for minority groups drops, caused by fewer available data points and the lower group ratio in the retain set.
>
>
> 2. We agree with the reviewer that saying “the more samples of a group are removed, the lower the resulting group accuracy” is inaccurate. What we meant is that by unlearning, group proportions in the retain set can diverge from the training set, due to the non-i.i.d. nature of real-world forget sets. If the proportion of a group in the retain set lowers after unlearning, then the resulting accuracy will be lower. Therefore, we rewrote the sentence above with *“As the proportion of a group in the retain set lowers, its resulting accuracy also decreases”* (see Section 3.2 of the revised paper).
>
>
> 3. We thank the reviewer for spotting this. Effectively disentangling the two dimensions (less data and different group proportions) is challenging, as, e.g., changing group proportions also reduces group data. We argue that both factors likely contribute to lowering the model’s generalization performance, with different intensities. We agree that the quoted sentence does not fully convey this message; therefore, we welcome any suggestions from the reviewer on how to improve this part of the claim.
>
> ### On the i.i.d. assumption of the forget set
> Indeed, existing works in the literature do allow for non-i.i.d. forget sets. Yet, their evaluation framework is based on the assumption that the forget set is i.i.d. in randomized unlearning. This assumption is also reflected in the methods’ designs for random unlearning, i.e., they do not account for group imbalance in the forget set distribution during unlearning. However, this does not prevent models from being used in non-i.i.d. settings, as we did in our evaluations. The aim of our work, and its primary contribution, is to raise awareness of this problem and propose a solution to prevent unbalanced distributions from harming the model’s robustness. Indeed, this objective adds another layer (i.e., preserving robustness) to the standard machine unlearning objective (i.e., minimize the discrepancy with the gold standard). We clarify this in Section 3.2.
>
> As a final note, class unlearning [2] it is a form of non-i.i.d. unlearning but has the opposite goal of our scenario as the final model should not be able to recognize a given target class after unlearning. Instead, our work addresses the problem of removing randomized data where the forget set is non-i.i.d., with the goal of preserving the original model performance on each group after unlearning.

---

> > ### Author Response · Authors · 2025-09-23
> >
> > ### Broader impact concerns
> > We acknowledge the reviewer's concerns about the nature of the attractiveness classification task. We originally picked “attractiveness” as the downstream task as previous works showed it has a great negative correlation with the protected attribute “male” [3, 4], i.e., it shows a Pearson Correlation of $\rho_{A, M}=-0.4$ against $\rho_{B, M}=-0.3$ of “blonde” and “male”. We understand that this can raise ethical concerns; therefore, we recomputed all CelebA experiments, substituting attractiveness as the target attribute with hair color (blond vs. non-blond). Furthermore, we removed every reference to the word “attractive” from the manuscript in the revised paper. MIU still shows the best performance overall. However, we notice that the task became easier, with smaller UA, EO, and GA discrepancies between the original and retrained model (without REWEIGHT). This results in smaller differences between different methods, with, e.g., MIU scoring an average gap of 99.0% vs 98.6% of SCRUB using an unlearning ratio of 0.5. As a final remark, we note that the broader impact statement is in the Conclusion section, but we called it “Negative societal impacts.” We updated the section for the revised paper by changing the paragraph name to “Broader impact statement.”
> >
> > ### Would strengthen the work
> > 1. As Section 4.1 (par. Datasets) details, we considered the worst-case scenario in which the forget set is sampled from a single group. Therefore, the unlearning ratio is defined as the proportion of samples from a particular group that have been unlearned. We clarified this in the revised paper (see Section 4.1).
> >
> > 2. While RWG and REWEIGHT share the same principle of reweighting the sampling distribution, they serve different purposes. RWG enforces a uniform sampling distribution, effectively maximizing the worst group accuracy. The proposed REWEIGHT, instead, changes the sampling distribution to match the original training set statistics. If the training set was originally balanced, then REWEIGHT behaves exactly as RWG. Thus, REWEIGHT can be seen as a generalization of RWG in the context of machine unlearning. Finally, we note that RWG may suffer the same issues of group-DRO as it targets the uniform sampling distribution. We clarified these aspects in Section 3.3.
> >
> > 3. We thank the reviewer for their suggestion; we moved the dominant forget-group accuracy (GA) definition to the main paper (see Section 4.1).
> >
> > 4. We thank the reviewer for suggesting these improvements. We updated the manuscript to address them.
> >
> > [1] Ruizhe Chen et al., Fast model debias with machine unlearning. In NeurIPS, 2024. \
> > [2] Kurmanji, Meghdad, et al. "Towards unbounded machine unlearning." NeurIPS, 2023. \
> > [3] Park Sungho et al., Fair contrastive learning for facial attribute classification. In CVPR, 2022. \
> > [4] Park, Sungho, and Hyeran Byun. "Fair-vpt: Fair visual prompt tuning for image classification." In CVPR, 2024.

---

> > > ### Author Response · Authors · 2025-10-02
> > >
> > > We deeply appreciate the reviewer’s time and effort in reviewing our paper. As the discussion window comes to an end, we hope our replies have addressed the concerns raised, and we would be glad to elaborate further on any aspects that may still be unclear.

---

> > > > ### Comment · Reviewer_xfyW · 2025-10-02
> > > >
> > > > Thank you to the authors for the comprehensive revision and discussion. I appreciate the extra experiments and my concerns are mostly satisfied. I have a couple short final comments:
> > > > * Regarding disentangling the data and group proportions: I agree this is challenging. I think the most scientifically rigorous way to study this would be a synthetic dataset wherein the data quantity, group ratios, etc. can be carefully controlled and ablated. However, given that this question is not central to the narrative, I would advise reducing any claims regarding group proportion being the primary causation of harm to generalization in situations where the absolute number of data is also reduced, and/or remarking on the status of data quantity as a confounding variable.
> > > > * Regarding the i.i.d. assumption on the forget set, I would appreciate a sentence or two of contextualization in the "machine unlearning" section of the related work in addition to the current revisions.

---

> > > > > ### Author Response · Authors · 2025-10-04
> > > > >
> > > > > We thank the reviewer for their positive feedback. Following their suggestions, we:
> > > > > - Updated the abstract, Section 3.2, and conclusion, reducing the statement on group proportion being the primary causation of harm to generalization.
> > > > > - Rewrote the second paragraph of "machine unlearning" in the related work, contextualizing the i.i.d. forget set assumption of existing works with group-robust machine unlearning.

---

### Review · Reviewer_vvqX · 2025-08-10

**Summary Of Contributions:**

The paper studies group-robust machine unlearning, where forget requests are uneven across demographic groups and can degrade accuracy for dominant groups; it formalizes this setting (assuming group labels) and considers both exact and approximate unlearning. It proposes (i) reweighting the retain-set sampling during retraining to preserve the original group robustness better than group-DRO, and (ii) MIU, an approximate method that minimizes mutual information between features and group labels with a calibration term; on CelebA, Waterbirds, and FairFace, these approaches better maintain group/test accuracy than prior unlearning methods.

**Audience:**

Yes

**Audience Explanation:**

Clearly, researchers from both machine unlearning and (group)-robust learning will be interested in this paper.

**Claims And Evidence:**

Yes

**Claims Explanation:**

This paper has a list of clear strengths:
- The problem is significant and novel, with a clear, well-motivated formulation.
- The paper is well-written and easy to follow.
- The method is intuitive, and the demo-code is provided for reproducibility.

However, potential weaknesses for this paper are:
- The method assumes access to oracle group labels, which is understandable for a first work, but should be complemented by discussion or simulations of mitigation strategies for missing group labels from the standard spurious-correlation literature.
- The evaluation is limited to vision datasets; adding text benchmarks (e.g., MultiNLI, CivilComments) would better demonstrate generalizability.
- The paper lacks a hyperparameter sensitivity analysis.

**Requested Changes:**

Aforementioned in the weakness part.

---

> ### Author Response · Authors · 2025-09-23
>
> ### On the access to oracle group labels
> Indeed, as the limitations paragraph outlines, ours is the first approach to tackle group-robustness in the machine unlearning field, and therefore, we assume access to group annotations. If group annotations are unavailable, existing works show that these can be discovered from data [1, 2]. According to Kim et al. (2024) [1], applying group-DRO to discovered group annotations closely matches the performance of group-DRO on ground-truth labels. Thus, future works designing group-agnostic methodologies could discover groups from data and use MIU and reweight as a baseline. We have updated the manuscript to include this discussion in the limitations (Section 5).
>
> ### Experiments on vision datasets
> As the reviewer correctly points out, we limited the evaluation to the image classification scenario, as outlined in the limitations paragraph. Since we are the first to address the non-i.i.d. nature of unlearning requests, we focused on image classification because machine unlearning is commonly evaluated in this task [3, 4, 5, 6]. Nonetheless, we updated the paper and included unlearning experiments on MultiNLI following the reviewer’s suggestion (see Section B.4). As Table 16 shows, reweight allows a successful recovery of the original group accuracy (-0.5), while MIU shows the greatest alignment with the gold standard both with and without reweight (96.0 and 95.8). These results confirm the findings already obtained for vision datasets.
>
> ### Hyperparameter sensitivity analysis
> Our work introduces two separate methodologies, i.e., REWEIGHT and MIU. The former does not introduce any hyperparameter, as it is a reweighting of the sampling distribution to match the training set statistics. In contrast, MIU only introduces the $\lambda$ hyperparameter, which effectively balances the contribution of the calibration term w.r.t. the retaining term (see the pseudocode). As Figure 8 in Appendix B.5 outlines, we ablated hyperparameter $\lambda$ by setting it to 0, 1, 5, and 10, showing that $\lambda=1$ generally yields the best average gap across the three datasets.
>
> [1] Younghyun Kim, Sangwoo Mo, Minkyu Kim, Kyungmin Lee, Jaeho Lee, and Jinwoo Shin. Discovering and mitigating visual biases through keyword explanation. In CVPR, 2024. \
> [2] Moreno D’Incà, Elia Peruzzo, Massimiliano Mancini, Dejia Xu, Vidit Goel, Xingqian Xu, Zhangyang Wang, Humphrey Shi, and Nicu Sebe. Openbias: Open-set bias detection in text-to-image generative models. In CVPR, 2024. \
> [3] Chundawat, Vikram S., et al. "Can bad teaching induce forgetting? unlearning in deep networks using an incompetent teacher." In AAAI, 2023. \
> [4] Fan, Chongyu, et al. "Challenging forgets: Unveiling the worst-case forget sets in machine unlearning." In ECCV, 2024. \
> [5] Jia, Jinghan, et al. "Model sparsity can simplify machine unlearning." In NeurIPS, 2023. \
> [6] Kurmanji, Meghdad, et al. "Towards unbounded machine unlearning." NeurIPS, 2023.

---

> > ### Author Response · Authors · 2025-10-02
> >
> > We are grateful to the reviewer once again for their valuable time and effort in assessing our work. As the discussion period draws to a close, we hope our response has sufficiently addressed the reviewer’s concerns. Should any points remain unclear, we would be pleased to offer further clarification.

---

### Review · Reviewer_KH27 · 2025-09-17

**Summary Of Contributions:**

The paper introduces a problem of group-robust machine unlearning. It first proposes to use a reweighting strategy to maintain robustness in exact unlearning (model retraining), and also introduces an approximate unlearning method based on mutual information minimization called MIU. The authors conduct experiments comparing MIU to prior works in unlearning on image classification datasets (CelebA, Waterbirds, FairFace). MIU achieves strong results in both preserving model generalization and closely matching exact unlearning model properties.

Overall, the proposed approximate unlearning method MIU is well-motivated and shows strong empirical performance. I also appreciate the presence of ablation studies on MIU.
However, I have a few concerns which would require a revision of the paper:
(1) clearly defining goals and the problem setting of machine unlearning and group-robust unlearning, (2) overall presentation clarity, and (3) over-claiming contribution of a novel exact unlearning method.

**Audience:**

Yes

**Audience Explanation:**

The paper proposes an interesting problem formulation where we want to maintain group robustness during machine unlearning. The proposed approximate unlearning algorithm is novel and claims around the method MIU are empirically supported. When the authors incorporate the suggested changes and address the concerns regarding the formulation and novelty of exact retraining, the paper will be a great addition to TMLR, for both machine unlearning and group robustness communities.

**Broader Impact Concerns:**

Ideally, if possible, change the CelebA attribute from “attractiveness” to a more neutral/academic choice like “smiling.” (see Requested Changes above)

**Claims And Evidence:**

No

**Claims Explanation:**

1. Issue with problem formulation of machine unlearning and group-robust machine unlearning setup.

a) For machine unlearning in Section 3.1: There are contradictions in whether the goal is (1) to forget $D_f$ while maintaining performance on the remain and test sets, or (2) to approximate the exact SGD retraining on the remain set as exactly as possible. These are two distinct goals, and this needs to be clarified.

b) In Section 3.2, for group-robust machine unlearning, similarly, is the goal to unlearn $D_f$ and (1) make the model as group robust as possible / maximum worst-group accuracy, (2) make it close to exact retraining with SGD on the remain set, (3) make it close to exact retraining with group robust learning on the remain set, or (4) minimize the accuracy losses on the dominant group in $D_f$, (5) make the accuracy of specific dominant group in $D_f$ close to the model with exact retraining …, etc? Please make this very clear in the problem statement.

2. The novelty of the exact retraining method is exaggerated.

The method described in Section 3.3 is not a novel contribution, and it is misleading to present it as such. Training with imbalance groups e.g. using reweighting is exactly what standard group robustness is doing, and it is not specific to unlearning [see Idrissi et al, Chaudhuri et al, among others]. If we just have $D_r$ with some group underrepresented, we know the strategist to get a group robust model with it from group robustness literature. I suggest that the authors present reweighting as a baseline for robust exact retraining but I don’t see a technical contribution there if I understand everything correctly.


[Idrissi et al https://arxiv.org/abs/2110.14503]

[Chaudhuri et al https://arxiv.org/abs/2205.11672]

**Requested Changes:**

1. (Most important) Improving clarity of the goals as discussed in “Issue with problem formulation of machine unlearning and group-robust machine unlearning setup.” above. Reframe the exact training method with reweighting as a baseline, not novel contribution, as discussed in “The novelty of the exact retraining method is exaggerated.” above.

Additional questions about the group-robust unlearning assumptions:
- Do you assume uniform or non uniform group distribution in the original train dataset?
- You assume that the forget set is skewed towards one group: but what if that group was overrepresented and now the retain set is balanced — does it change anything for the problem formulation? Isn’t it more important what group distribution of the retain set is, than what group distribution the forget set has?
- Do you assume that your original Pretrain model is group robust or not (i.e. is it trained with SGD, reweighting or GDRO)? E.g., in Section 3.4, page 5 “While preserving original group robustness” sounds like you do assume the original model is group robust?


2. Retraining baseline: the argument about Group DRO inflating forget set accuracy is confusing, since retraining and original training should align in methodology. It would be helpful to have baselines of Pretrain and Retrain models which align in training strategy. Specifically, have a Pretrain and Retrain pair both trained with SGD, GDRO, or reweighting.

The paper doesn’t have line numbers, so I’ll copy over a paragraph which was confusing to me:
“As a byproduct of strongly optimizing worst-group accuracies, we notice that group-DRO (Sagawa et al., 2020) can also increase the forget set accuracy. This issue makes approximate unlearning evaluation more difficult if Retrain + group-DRO is used as the gold standard. Assuming a hypothetical original forget-set accuracy of 70%, if an approximate unlearning algorithm targeting such a gold standard leads to higher accuracy (e.g., 80%), then was the knowledge unlearned if forget-set accuracy increased?”

Please clarify this, since it sounds like GDRO leads to better generalization, and not using an algorithm which leads to better generalization / group robustness seems really confusing.

3. Overall clarity, questions and suggested changes:
* Please add error bars to the reported results.
* Provide full experimental context for Figures 2, 3: training setup, data splits, target, models, group distributions, forget set details. What does y-axes shoes? Can you add group accuracies / average accuracy / WGA? What’s the difference between left and right subplot in Fig 3?
* In Experiments section, which groups do you use for froget set?
* Generally report worst-group accuracy (WGA).
* Define all metrics / terms clearly: e.g., “average gap”, “unlearning ratio” (wrt group size, forget set size, or full dataset?).
* Eq. (2): define $T_{\psi}$.
* Section 4.4: report group distribution and size of forget set.
* typo: “This section describes the experimental protocol (Sec. 3.1)” -> should be 4.1.
* Figures/Tables: indicate which group forms the unlearning set.
* Ideally, if possible, change the CelebA attribute from “attractiveness” to a more neutral/academic choice like “smiling.”
* Please clarify this claim: “Intuitively, increasing the sampling likelihood for partly unlearned groups rebalances the retain set group statistics to match those of the training dataset.”
* Please elaborate on this observation at page 11: “Instead, all algorithms struggle at 0.1 unlearning ratio in Waterbirds (Sagawa et al., 2020), where the small forget set size causes UA fluctuations and affects the BatchNorm (Ioffe, 2015) estimation”.
* Section 4.4: what is the group distribution and size of the forget set?

---

> ### Author Response · Authors · 2025-09-23
>
> ### On the MU and Group-robust MU formulation
> We thank the reviewer for the constructive feedback. We updated the paper as follows: \
>     a) The main goal of machine unlearning is to approximate the model trained from scratch on the remaining set (i.e., without $D_f$ ) as closely as possible. To remove inconsistencies, we revised Section 3.1 removing the first inconsistent statement and adjusting the second as: “*a machine unlearning algorithm $\mathcal{U}$ scrubs the influence of a desired forget set $D_f \subset D_{tr}$ from the pretrained model by outputting scrubbed weights $\{\varphi_u, \theta_u\}$ such that $h_{\varphi_u} \circ f_{\theta_u}$ is as close as possible to the exact unlearning model $h_{\varphi_r}\circ f_{\theta_r}$ (or Retrain), trained solely on the remaining set $D_r = D_{tr} \setminus D_f$ with algorithm $\mathcal{T}$.*” \
>     b) The main difference between standard machine unlearning and group-robust machine unlearning is that the latter aims to preserve the original model’s robustness, something that a non-uniform forget set may harm. To clarify this, we appended a new paragraph to Section 3.2 stating that “*the objective of group-robust machine unlearning differs from that of machine unlearning. While the latter minimizes the discrepancy with the retrained model (gold standard), the former also
> preserves the original model’s performance on each group.*”
>
> ### On our contribution for REWEIGHT
> We acknowledge the reviewer's concern and have updated Section 3.3 of the manuscript to highlight the differences between our reweighting strategy and the one commonly used in the group-robustness literature. REWEIGHT is an adaptation of the approach commonly used in the literature in order to fulfill the goal of group-robust machine unlearning. In fact, the key distinction lies in how the distribution is reweighted. In group-robust optimization, the goal is to maximize the worst-group accuracy; thus, reweighting the sampling distribution is used to produce uniformly distributed samples across groups [1]. In contrast, our goal is to maintain the robustness of the original model after unlearning: thus, REWEIGHT aligns the sampling distribution with that of the original dataset, mimicking the training of the original model.  We clarified this in Section 3.3. By stating “*Contrary to RWG (Idrissi et al., 2022), which enforces uniform sampling distribution from group-perspective, we suggest instead to reweight the sampling distribution to match the original dataset statistics.*”.
>
> [1] Idrissi, Badr Youbi, et al. "Simple data balancing achieves competitive worst-group-accuracy." Conference on Causal Learning and Reasoning. PMLR, 2022.

---

> ### Author Response · Authors · 2025-09-23
>
> ### Requested Changes
> 1. See above (**On the MU and Group-robust MU formulation**)
>
>     1.1. We do not make assumptions about the training set distribution. Yet, we employed commonly used datasets in the group-robust optimization literature, which are non-uniformly distributed.
>
>     1.2. We thank the reviewer for raising this point. As also pointed out by reviewer xfyW, Section 3.2 inaccurately states that “the more samples of a group are removed, the lower the resulting group accuracy.” The meaning behind the statement was that, if the proportion of a group in the retain set lowers after unlearning compared to the training set, then the accuracy of the model for such a group will decrease. Therefore, we revised Section 3.2 by improving the phrasing and showing (Figure 2) how accuracy changes when unlearning a minority and a majority group.
>
>     1.3. We do not assume that the original model is trained with a group-robust optimization approach, therefore we train the model via empirical risk minimization. We clarified this in the revised paper, Section 4.1.
>
> 2. We agree with the reviewer that retraining and original training should align in methodology. However, the goal of group-robust machine unlearning is to preserve the degree of robustness of the original model *no matter the original training strategy*. Thus, in the paper we consider a widely used pretraining strategy, i.e., via empirical risk minimization (ERM).  In this context, if we use Group DRO to retrain a model pretrained with ERM, it could inflate the forget set accuracy, even beyond its original value (e.g., UA in Table 1 and 2).  If an approximate unlearning method tries to mimic such behaviour, it would also end up increasing the forget set accuracy, creating a paradoxical situation.
>
> 3. Overall clarity, questions and suggested changes
>
>     3.1 We avoided including error bars in plots, as they would make the plot harder to read. Furthermore, for each plot, we included the associated table(s) with standard deviations either in the main paper or in the supplementary material.
>
>     3.2 As we updated Figure 2, we list these details for the updated plot. We use a ResNet-18 trained with ERM on CelebA as the original model. We employ different methods to jointly unlearn varying proportions of blond males and non-blond females, showing how the test accuracy varies for these groups (blond males on the left and non-blond females on the right). Therefore, the y-axis represents the accuracy on the subset of the test set composed of these groups. The two plots in Figure 3 present different issues that we encountered when using group-DRO for model retraining, namely, an increase in the accuracy of the dominant group of the forget set (see GA in Table 1 and in Table 2) and a drop in the test set accuracy (see TA in Table 3). The plot measures the inverse gap with the original model; therefore, the higher, the better.
>
>     3.3 As we describe in Section 4.1, groups are blonde males for CelebA, waterbirds on land for Waterbirds, and 20-29 y.o. afro-americans for FairFace, where blonde, waterbirds, and 20-29 y.o. are the target attributes.
>
>     3.4 We thank the reviewer for the suggestion. As we already have numerous metrics, and we are more interested in the accuracy drop for the dominant forget set accuracy (i.e., the group that is the target for unlearning) and the overall model fairness, we only reported WG accuracy in the Appendix.
>
>     3.5 Each metric (except for GA) is described in Appendix A. Furthermore, we updated Section 4.1, explaining what the unlearning ratio represents.
>
>     3.6 $T_{\psi}$ is a simple 2-layer MLP with ReLU activation that outputs a scalar. We clarified this in Section 3.4.
>
>     3.7 We updated the paper and reported the forget set composition for the Section 4.4 experiment.
>
>     3.8 We thank the reviewer for spotting this typo.
>
>     3.9 As groups are already described in Section 4.1, we avoided redundant explanations about group compositions.
>
>     3.10 As also suggested by reviewer xfyW, we changed “attractive” to “blond” as the target attribute for CelebA experiments.
>
>     3.11 In other terms, the proposed reweighting strategy tries to recreate the original training conditions, net of fewer samples. Therefore, it rebalances the sampling distribution, ensuring the probability of sampling a group is the same as the original training dataset.
>
>     3.12 When unlearning 10% of waterbirds on land, it means unlearning five samples. This leads to a couple of issues. First, the forget set accuracy (UA) fluctuates as each correctly predicted sample raises the UA by 20% (1/5). Then, both gradient estimation and BatchNorm estimation are extremely noisy as they are averaged over five samples. This explains why an unlearning ratio of 0.1 is harder than 0.5 and 0.9 in Waterbirds. We clarified this in Section 4.3.
>
>     3.13 See 3.7.

---

> > ### Author Response · Authors · 2025-10-02
> >
> > We sincerely thank the reviewer once again for their time and effort in evaluating our paper. As the time for discussion is closing, we hope our response has adequately addressed the reviewer’s concerns. If any part remains unclear, we would be glad to provide further clarification.

---

### Author Response · Authors · 2025-09-23

### General Comment
We sincerely thank all reviewers and the action editor for their time and effort in evaluating our work. We are encouraged that the reviewers found group-robust unlearning to be an interesting (`KH27`), novel (`vvqX`, `xfyW`), and significant (`vvqX`, `xfyW`) problem for the research community. We particularly appreciate the recognition that our work “has the potential to bring two research communities together” (`xfyW`). Regarding our proposed approximate unlearning method (`MIU`), we are pleased that reviewers found it to be novel (`KH27`), clearly presented (`xfyW`), and well motivated (`KH27`).

### Summary of the Rebuttal
We appreciated the constructive suggestions proposed by reviewers and updated the manuscript (changes in blue) by:
1. Recomputing CelebA experiments using “blond” as the target attribute (also adjusting the results description accordingly).
2. Reporting results on the natural language dataset MultiNLI (Section B.4)
3. Clarifying the machine unlearning and group-robust machine unlearning formulation (Sections 3.1 and 3.2).
4. Expanding the experiments of Figure 2 and Figure 7 by showing accuracy degradation for minority and majority groups.
5. Fixing the claim that “The more samples of a group are removed, the lower the resulting group accuracy. [...]” in Section 3.2.
6. Compared the proposed REWEIGHT with the data distribution reweighting strategy commonly used in group-robust optimization (Section 3.3).
7. Providing a formal definition of the unlearning ratio in group-robust machine unlearning (Section 4.1).
8. Moving the definition of GA to the main paper (Section 4.1).
9. Discussing a possible approach for group-robust machine unlearning in the absence of group annotations.

---

### Decision · Action_Editor_87pm · 2025-11-11

**Recommendation:** Accept as is

**Audience:**

Yes

**Audience Explanation:**

The paper bridges the fields of empirical group robustness (mostly in the context of spurious correlations) and machine unlearning and is likely to be of interest to both communities.

**Claims And Evidence:**

Yes

**Claims Explanation:**

All the reviewers were satisfied by the claims made in the paper after the revision process. Some broader impact concerns were raised with the attractiveness label in CelebA in initial experiments, but the authors modified this experiment during the revision phase.